# Increased reproducibility of brain organoids through controlled fluid dynamics

Giuseppe Aiello [ID]1, Mohamed Nemir [ID]1,6, Barbora Vidimova1,6, Cindy Ramel1, Joanna Viguie1, Arianna Ravera2, Krzysztof Wrzesinski [ID]3,4 & Claudia Bagni [ID]1,5 ✉

## Abstract

Brain organoids are a promising model for studying human neurodevelopment and disease. Despite the potential, their 3D structure exhibits high variability during differentiation across batches and cell lines, presenting a significant challenge for biomedical applications. During development, organoids are exposed to fluid flow shear stress (fFSS) generated by the flow of culture media over the developing tissue. This stress is thought to disrupt cellular integrity and morphogenesis, leading to variation in organoids architecture, ultimately affecting reproducibility. Understanding the interplay between tissue morphology, cell identity and organoid development is therefore essential for advancing the use of brain organoids. Here, we demonstrate that reducing fFSS, by employing a vertically rotating chamber during neuronal induction, a critical phase for organoid morphogenesis, along with an extended cell aggregation phase to minimize fusions, significantly improves the reproducibility of brain organoids. Remarkably, reducing fFSS minimizes morphological structure variation and preserves transcriptional signature fidelity across differentiation batches and cell lines. This approach could enhance the reliability of brain organoid models, with important implications for neurodevelopmental research and preclinical studies.

Keywords Fluid Flow Shear Stress (fFSS); Neurodevelopment; Morphological Reproducibility; Metabolism
Subject Categories Methods & Resources; Neuroscience

## Introduction

Brain organoids have become invaluable tools for advancing our understanding of human neurodevelopment, modeling neurological disorders and exploring personalized medicine. These models, compared to classic differentiation models in monolayer cultures, have shown higher complexity and similarity to the developing human brain, offering an insight into processes that were previously inaccessible (Antonica et al, 2022; Camp et al, 2015; Eichmuller and Knoblich, 2022; Kelley and Pasca, 2022; Kim et al, 2020; Lancaster et al, 2013; Luo et al, 2016; Pellegrini et al, 2020; Qian et al, 2019).

A key strength of brain organoids lies in their ability to mimic the intricate relationship between tissue structure and gene expression, where precise morphological organization underpins the tissue's functional complexity (Camp et al, 2015; Fleck et al, 2023; Luo et al, 2016; Quadrato et al, 2017; Warmflash et al, 2014). However, brain organoid technology faces a critical barrier to widespread adoption related to reproducibility (Di Lullo and Kriegstein, 2017; Qian et al, 2019). Variability in tissue morphology, cellular composition and differentiation outcomes across batches and cell lines limits their reliability. This inconsistency poses significant challenges for comparative studies and high-throughput applications, reducing their utility in both basic and translational research (Bhaduri et al, 2020; Lancaster et al, 2017; Messana et al, 2008; Velasco et al, 2019).

Tissue morphology is increasingly recognized as a critical determinant of organoid development, and recent studies have highlighted its importance in guiding the temporal and spatial aspects of this process (Gjorevski et al, 2022; Lancaster et al, 2017). Morphological inconsistencies not only affect the structural fidelity of organoids but also alter the temporal program of differentiation, impacting regional identity, cellular organization and functional properties (Chiaradia et al, 2023; Jain et al, 2025; Martins-Costa et al, 2023).

Extracellular matrices (ECM), such as solubilized basement membrane Matrigel, are commonly used and are known to impact organoid morphology, potentially influencing organoid development (Jain et al, 2025; Long et al, 2018; Martins-Costa et al, 2023; Pagliaro et al, 2025). The complex biochemical composition of Matrigel remains undefined and difficult to standardize across batches due to its extraction from Engelbreth–Holm–Swarm mouse sarcomas (Aisenbrey and Murphy, 2020). Despite facilitating initial morphological organization and neuroepithelial formation, this undefined ECM may affect endogenous ECM production, contributing to variability in differentiation outcomes across experiments (Jain et al, 2025; Long et al, 2018; Martins-Costa et al, 2023; Pagliaro et al, 2025).

1Department of Fundamental Neurosciences, University of Lausanne, Lausanne, Switzerland. 2Scientific Computing and Research Support Unit, University of Lausanne, Lausanne, Switzerland. 3Centre of Excellence for Pharmaceutical Sciences (Pharmacen™), North-West University, Potchefstroom, South Africa. 4CelVivo ApS, Odense, Denmark. 5Department of Biomedicine and Prevention, University of Rome Tor Vergata, Rome, Italy. 6These authors contributed equally: Mohamed Nemir, Barbora Vidimova. ✉E-mail: Claudia.Bagni@unil.ch

These insights underscore the need for methods that standardize the morphological aspects of brain organoid development, as irregularities in tissue structure can propagate through later stages of development, contributing to variable phenotypical outcomes. While much attention has been given to intrinsic factors, such as genetic variability between cell lines, the influence of extrinsic biomechanical factors is often underestimated.

In the developing embryo, mechanical forces play a fundamental role in shaping tissues and guiding differentiation: contractility and tension create a mechano-transduction feedback loop that regulates morphogenesis and patterning, influencing gene expression through pathways such as SRC-dependent β-catenin signaling (Caldarelli et al, 2024). This highlights the intricate crosstalk between mechanical forces and molecular pathways as key regulators of development.

Brain organoids, despite lacking maternal cues and extraembryonic tissues, are highly sensitive to external mechanical perturbations, particularly during the earliest stages of differentiation. The initial aggregation and the subsequent neuronal induction phases are critical windows where tissue identity and patterning are established. During these stages, cells undergo self-organization, lineage specification and neuroepithelium formation, processes that depend on finely balanced mechanical cues (Andrews and Nowakowski, 2019; Renner et al, 2017).

In the development of brain organoids, extrinsic force variation can disrupt tissue integrity and morphogenesis, leading to organoids that differ in shape and functionality (Goto-Silva et al, 2019; Ishihara et al, 2023; Saglam-Metiner et al, 2023). One such factor is fluid flow shear stress (fFSS), which arises from the flow of culture media in dynamic systems commonly used for organoid culture (Dahl-Jensen and Grapin-Botton, 2017; Goto-Silva et al, 2019; Hinton et al, 2022). These systems, including plates, bioreactors and spinning bioreactors, are designed to improve nutrient and oxygen distribution, promoting cell survival and growth. However, the trade-off is the generation of fFSS, a mechanical force that can disrupt cellular integrity and tissue organization (Croughan and Wang, 1991; Goto-Silva et al, 2019; Suong et al, 2021). Prolonged exposure to fFSS during critical stages of organoid development has been shown to interfere with proper morphogenesis, leading to inconsistencies in tissue architecture that ultimately affect reproducibility (Dardik et al, 2005; Gareau et al, 2014; Ismadi et al, 2014; Saglam-Metiner et al, 2023).

Additionally, cellular metabolism is increasingly recognized as a key driver of neurodevelopment, influencing energy states, signaling pathways and overall tissue health (Badal et al, 2019; Iwata et al, 2023; Iwata et al, 2020; Khacho and Slack, 2018). Metabolic heterogeneity in brain organoids is tightly linked to differentiation outcomes, with variations in oxidative phosphorylation (OXPHOS) and glycolysis influencing progenitor maintenance and neuronal specification (Øhlenschlæger et al, 2023). Nonetheless, it remains unclear whether changes in metabolic states across conditions in organoids are linked to variability in cellular composition or genuinely driven by biological differences.

Here we demonstrate that reducing fFSS during neuronal induction, a critical phase of cortical patterning, significantly improves the morphological reproducibility of brain organoids. By employing a rotating chamber (RC) apparatus, designed to minimize fFSS, and using an ECM-free approach for greater control over differentiation conditions, we achieved consistency of morphological features and transcriptional fidelity across organoid batches and cell lines. Furthermore, our transcriptional analysis revealed that RC organoids exhibit a higher expression of OXPHOS-related genes compared to glycolysis-related genes, suggesting a metabolic shift towards oxidative phosphorylation. To further characterize these metabolic states, we developed a novel protocol, which enables precise and reliable quantification of mitochondrial oxygen consumption (OCR). Our findings emphasize the importance of controlling the physical culture environment to ensure consistent outcomes in human neurodevelopmental models while showcasing the impact of our newly established culturing strategy on implementing state-of-the-art technologies.

## Results

### Near-microgravity conditions for the generation of brain organoids

To minimize the mechanical stress during key stages of differentiation, we made use of a Clinoreactor system (https://celvivo.com), herein called Rotating Chamber (RC), which enables precise control over fluid dynamics during organoid development. Unlike traditional bioreactors or spinning platforms that expose the developing tissues to fFSS, the RC utilizes gentle rotational dynamics to achieve near-microgravity conditions, reducing shear stress while maintaining optimal nutrient and oxygen exchange (Wrzesinski and Fey, 2018) (Fig. 1A). Computational fluid dynamics (CFD) analysis performed for the RC demonstrated an even distribution of shear forces within the culture chamber (Methods, Fig. 1B). Shear forces remained consistently low (<14 mPa), a range suitable for promoting organoid development without causing mechanical disruption (Saglam-Metiner et al, 2023; Velasco et al, 2020). Notably, these values are much lower than the shear forces normally present in an orbital shaker culture system, and range between 1.57 and $1.93 \times 10^{-2}$ Pa for velocities between 0.42 and $7.24 \times 10^{-2}$ m/s (Saglam-Metiner et al, 2023; Wrzesinski and Fey, 2018). Additionally, velocity and shear stress simulations confirmed stable flow dynamics across constructs with different densities (defined as relative weight/buoyancy of structures within the same surface area) (Fig. 1C; Movie EV1 and Methods). The shear stress levels remained low across all tested conditions with rotation speed ranging from 3 to 26 RPM, depending on the construct size and density (relative weight/buoyancy), confirming the RC's ability to maintain a reproducible and mechanically stable culture environment (Fig. 1D).

### Differentiation of brain organoids in RC conditions

To generate brain organoids, we adapted a previously published protocol for dorsal forebrain differentiation (Khan et al, 2020; Sloan et al, 2018) (Methods, Fig. 2A, top). This protocol was chosen as it is matrix-free, and therefore aligns with our objective to limit the variability and presence of undefined factors associated with ECM-based products (Jain et al, 2025; Long et al, 2018; Martins-Costa et al, 2023).

First, we generated embryoid bodies (EBs) from hiPSCs and extended the initial aggregation phase to 6 days, with minimal

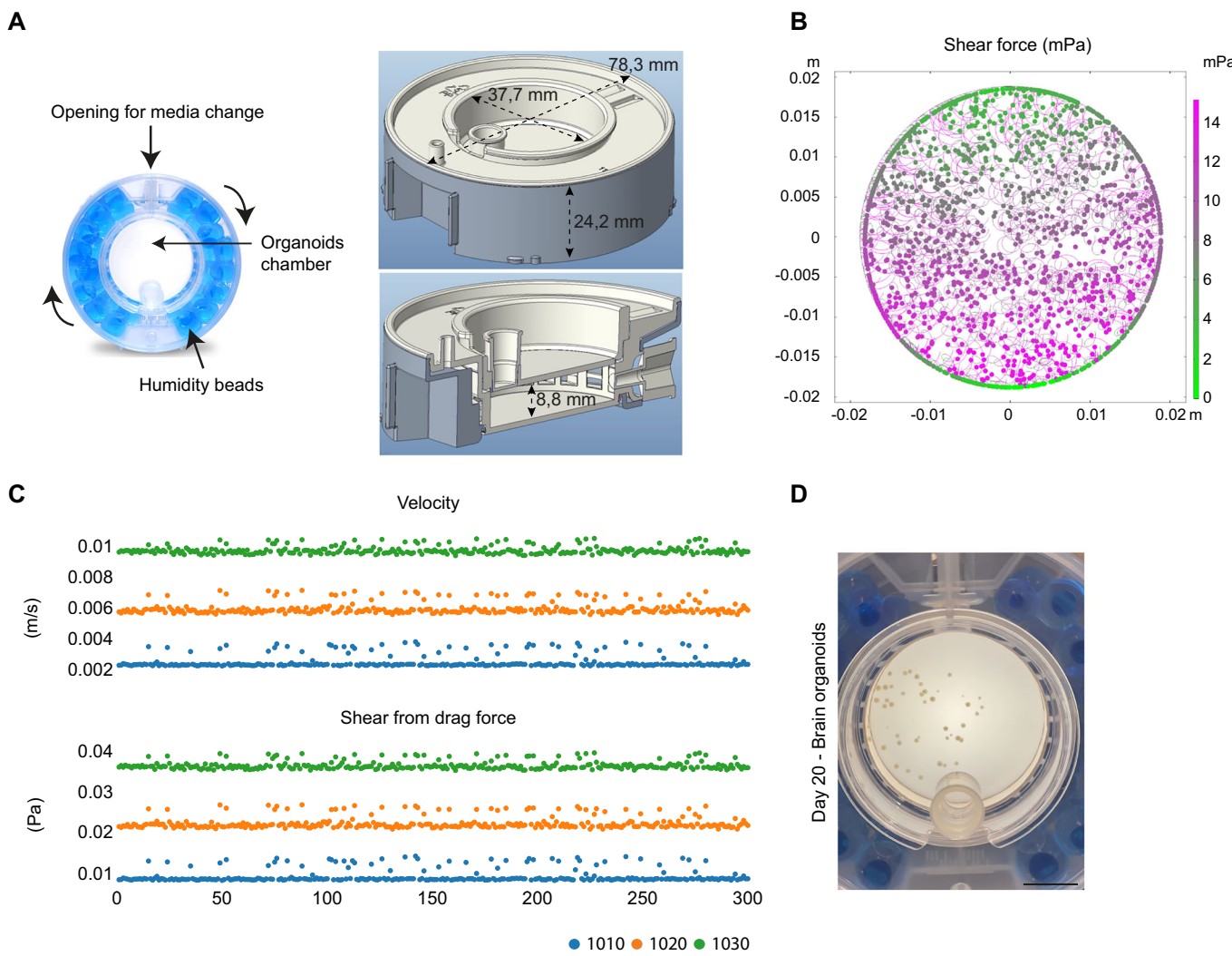

**Figure 1. Regulation of shear force affects organoid development.**

(A) Schematic representation of the rotating chamber (RC), illustrating its rotational dynamics designed to minimize fluid flow shear stress (fFSS) while ensuring uniform nutrient and oxygen distribution. The representation highlights key features, including the opening for media exchange, the organoid culture chamber, and the surrounding humidity beads to maintain optimal culture conditions. The right panels provide detailed dimensions of the RC apparatus, with an outer diameter of 78.3 mm and a height of 24.2 mm. The inner culture chamber has a diameter of 37.7 mm and a height of 8.8 mm, providing ample space for organoid growth. (B) Computational fluid dynamics (CFD) (COMSOL Multiphysics 6.2) simulation depicting the distribution of shear forces within the culture chamber. The shear force map reveals consistently low values (<20 mPa), indicating a mechanically stable environment conducive to organoid development. (C) Velocity and shear stress analysis for constructs with different densities (Generated in COMSOL Multiphysics 6.2 by Resolvent Denmark PS, Maaloev, Denmark; relative mass 1010, 1020, and 1030, where 1000 is the density/relative mass of the surrounding medium). The top graph shows the velocity distribution, while the bottom graph illustrates shear stress from the drag force. Both analyses confirm the stability and reproducibility of the RC system in maintaining low-shear conditions for all individual objects within the organoid chamber (objects 0 to 300). (D) Representative brightfield image of brain organoids cultured in the RC system at day 20. Scale bar: 1 mm. Source data are available online for this figure.

disturbance, in ultra-low-attachment microwell culture plates (AggreWell™) to reduce the risk of early fusion events. The EBs were then transferred into the RC apparatus for the entire neural induction phase (days 6–25) in the presence of EGF and FGF2. Within the RC system, the organoids remained gently suspended in the chamber, and exhibited a smooth, uniform, and spherical morphology (Fig. 2A; Movie EV2).

For the differentiation phase (days 25–45), organoids were transferred to low-attachment tissue culture plates maintained on an orbital shaker, in the presence of brain-derived neurotrophic factor (BDNF), neurotrophin-3 (NT-3), cyclic AMP (cAMP), L-

ascorbic acid (L-AA), and docosahexaenoic acid (DHA) (Methods). Thereafter (days 45–90), organoids were kept in maturation medium (Neurobasal A, B-27 Plus). Morphologically, organoids transitioned from compact EBs at day 2 to more organized and enlarged structures by day 6 (Fig. 2A bottom). As differentiation progressed, concomitant with a steady growth in size (Fig. 2B), the organoids maintained a spherical-like shape, which is critical for uniform development of the different organoids within the same batch, consistent with ongoing proliferation and maturation (Fig. 2A bottom). Organoid area increased significantly over time, with the largest values observed at day 90 (Fig. 2B).

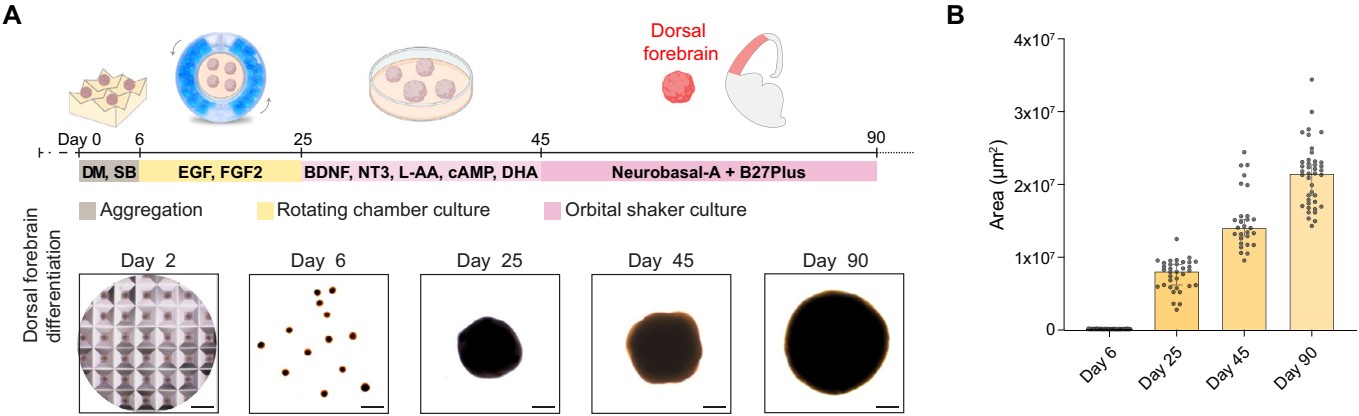

**Figure 2. Generation of brain organoids with the RC system.**

(A) Schematic representation of the experimental timeline for RC organoid generation, from aggregation (day 0) to maturation (day 90). Representative brightfield images illustrate the morphological progression of dorsal forebrain organoids at different time points. Scale bars: 500 μm. (B) Quantification of organoid areas at different time points, showing a significant increase in size from day 6 to day 90. Data were presented as mean ± SEM. Source data are available online for this figure.

## RC organoids exhibit greater morphological consistency across different batches and cell lines

Morphological parameters are critical indicators of organoid health, quality, reproducibility and functionality (Chiaradia et al, 2023; Lancaster et al, 2017; Øhlenschlæger et al, 2023). We developed an automated Python-based pipeline to extract and quantify 10 distinct, previously established morphological parameters (Chiaradia et al, 2023) (Fig. 3A and Methods). This approach enables objective, scalable, and reproducible morphometric analysis, minimizing user bias and ensuring consistency across experiments. We conducted this analysis at two different stages of organoid development, days 6 and 90, comparing two alternative culture protocols across two different hiPSC lines.

First, we evaluated the impact of the methodology of aggregation on embryoid bodies (EBs) morphology (Fig. 3B top). We compared short (PA protocol, 1 day) and long (RC protocol, 6 days) aggregation using AggreWell™ plates using the same media formulation (Methods). Therefore, the PA protocol involves the short aggregation in AggreWell™ plates for the first day, followed by 5 days of culture in low attachment 10 cm plates, as described in Khan et al, 2020; Sloan et al, 2018, while the RC protocol consists of 6 days of aggregation in AggreWell™ plates. At day 6, brightfield images revealed considerable differences in the morphology of PA and RC EBs (Fig. 3B bottom). EBs grown with the PA protocol tend to fuse, resulting in cellular aggregates of various sizes and shapes. By contrast, EBs grown with the RC protocol exhibit a spherical-like morphology, a more uniform size, and rarely fuse together. Principal component analysis (PCA) of all the morphometric measurements shows that PA EBs exhibit a larger confidence ellipse than RC EBs (Fig. 3C), indicating greater morphological heterogeneity. The contribution of each morphological parameter, calculated as the cosine² across the principal components (PCs) of the PCA, is summarized in a correlation plot (Fig. 3D). Each parameter contributes strongly to the first two dimensions of the PCA, except for roundness, which is represented to a lesser extent, and the mean curvature, which is better characterized by the PC3. Except for the perimeter, all other morphological parameters are

significantly different between the two protocols (Fig. EV1A). Overall, RC EBs are larger than PA EBs, with statistically significantly higher median areas and average radii. Similar observations are noted for the minimum and maximum Feret diameters. Importantly, the median roundness values for PA and RC EBs are 85.5 and 96.9%, respectively, indicating that the RC protocol generates more spherical EBs (Fig. EV1A; Dataset EV1). In addition, PA EBs display a significantly wider distribution across all morphological variables (Dataset EV2).

To assess the inter-batch variability, we quantified the absolute value of the relative median absolute deviation (MAD) (Fig. EV1B; Dataset EV3). Except for the standard curvature and the Dirichlet normal energy (DNE), all absolute relative MADs indicate a significant difference in variability across batches, with the PA protocol displaying a greater value of MAD (Mann–Whitney test, $p < 0.05$). Taken together, these results suggest that the RC EBs demonstrate higher inter-batch morphological reproducibility compared to the PA EBs.

To further investigate the reproducibility of morphological features across different cell lines, we analysed the inter-line variability within the RC and PA protocols. RC EBs exhibit a more consistent distribution of morphological measurements across both lines, with no significant differences in area, perimeter, or average radius (Fig. EV1C). By contrast, PA EBs display greater differences between the two lines, particularly in roundness and curvature-related parameters. The median roundness values show a more pronounced difference between the two lines in the PA protocol compared to the RC protocol, indicating that the RC system facilitates greater morphological uniformity across cell lines with different genetic backgrounds. The inter-line variability was further quantified through standard curvature and DNE measurements (see Methods), which showed strong morphological reproducibility within lines only for the RC protocol, while the PA protocol produced highly variable values in both lines, resulting in no statistically significant difference (Fig. EV1C). This indicates that while the RC protocol enhances inter-batch reproducibility, subtle morphological differences across cell lines may persist due to inherent genetic variability. The results suggest that the RC

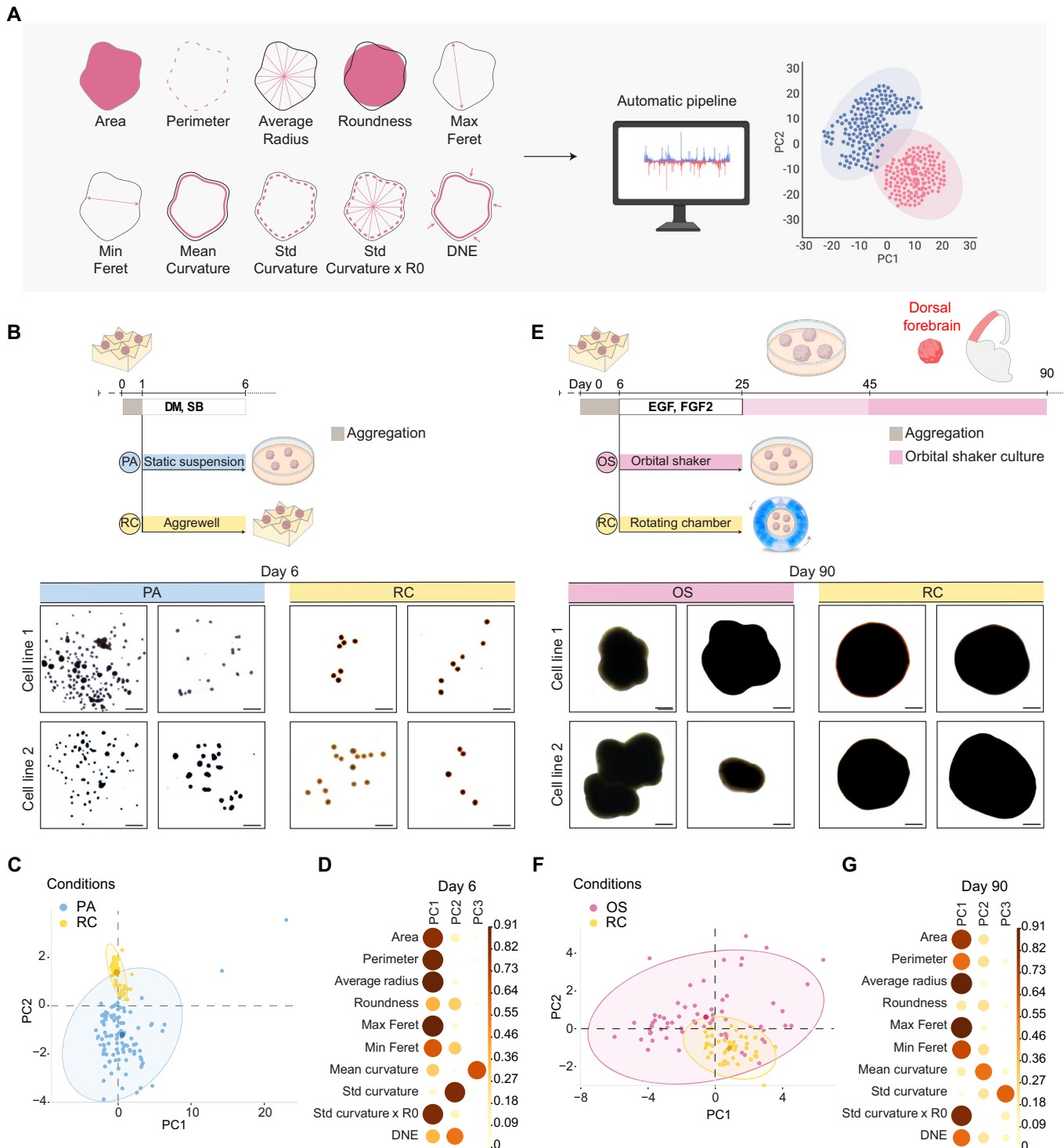

protocol not only enhances inter-batch reproducibility, but also mitigates morphological variability across cell lines with different genetic backgrounds.

As a second comparison, we focused on the impact of the RC versus the standard orbital shaker (OS) culture conditions. We followed the same differentiation protocol for both conditions, except that, during neural induction (days 6 to 25), EBs were cultured either in the RC system or using the OS-based method (Fig. 3E top). Organoids were generated from two different hiPSC lines and harvested for analysis at a more mature stage (day 90). OS organoids from both lines exhibit irregular and multilobate morphology. Conversely, RC organoids display consistently more regular and spherical shapes (Fig. 3E bottom). Considering all morphometric parameters in the PCA analysis, OS organoids

Figure 3.   RC organoids exhibit reduced morphological variability across batches and cell lines.

(A) Overview of the automated morphological analysis pipeline. Ten morphological parameters, including area, perimeter, average radius, roundness, Feret diameters, curvature metrics, and Dirichlet normal energy (DNE), were extracted from organoid images. The dataset was processed using an automated Python-based code, and principal component analysis (PCA) was applied to assess variability across conditions. (B) Top: Schematic representation of the PA and RC protocols, highlighting key differences in culture conditions. PA organoids are aggregated for 1 day and cultured in static conditions, while RC organoids undergo a six-day aggregation phase in an AggreWell™ plate. Bottom: Representative brightfield images of embryoid bodies (EBs) at day 6, comparing PA and RC protocols across two independent iPSC lines. Each picture represents EBs of different batches. PA EBs display a heterogeneous morphology with frequent aggregation, while RC EBs exhibit a more uniform, spherical shape. Scale bars: 500 μm. (C) PCA analysis of morphological parameters at day 6, comparing PA (blue) (n = 102, five batches) and RC (yellow) (n = 88, three batches) conditions. The confidence ellipses englobe 95% of the EBs within the same condition and its center, indicating the mean of all the data, is highlighted with a darker color. (D) Correlation plot showing the contribution of ten morphological parameters to the first three dimensions of PCA at day 6. The importance of each parameter is represented as cosine² values across the principal components. The size and color intensity of the circles are proportional to the parameters' contribution to the principal components. (E) Top: Schematic representation of the OS and RC protocols, highlighting key differences in culture conditions. From days 6 to 25, OS organoids are maintained in a 10 cm plate on an orbital shaker, while RC organoids are cultured in a rotating chamber. Bottom: Representative brightfield images of organoids at day 90, comparing OS and RC protocols across two iPSC lines. Each picture represents organoids from different batches. OS organoids exhibit irregular and lobulated morphologies, whereas RC organoids maintain a more uniform, spherical shape. Scale bars: 500 μm. (F) PCA analysis of morphological parameters at day 90, comparing OS (pink) (n = 67, five batches) and RC (yellow) (n = 41, five batches) conditions. Each dot represents one organoid. The confidence ellipses englobe 95% of the organoids within the same condition and its center, indicating the mean of all the data, is highlighted with a darker color. (G) Correlation plot showing the contribution of ten morphological parameters to the first three dimensions of the PCA at day 90. The importance of each parameter is represented as cosine² values across the principal components. The size and color intensity of the circles are proportional to the parameters' contribution to the principal components. Source data are available online for this figure.

present a broader ellipse compared to RC organoids (Fig. 3F), indicating greater morphological heterogeneity. The contribution of each morphological parameter reveals that the features are well represented by the first two dimensions of the corresponding PCA plot, except for roundness and standard curvature (Fig. 3G). This is further illustrated in the bar plots of each variable (Fig. EV2A). Notably, the median roundness values of OS and RC organoids are 78.0 and 90.2%, respectively. Among the measured morphological parameters, area, average radius, minimum and maximum Feret diameters significantly differ between the two groups, indicating that the RC protocol tends to generate larger organoids. Furthermore, the OS protocol produces organoids with a wider distribution of their morphometric values compared to RC organoids for all features, except for roundness and standard curvature (Dataset EV2). The inter-batch variability is illustrated with the absolute relative MAD (Fig. EV2B; Dataset EV4). OS batches show greater variability in perimeter and roundness values compared to RC batches. These findings suggest that the RC protocol allows the generation of morphologically more reproducible organoids, even during long-term culture across different cell lines. To further assess the positive effect of controlling the shear force and fluid dynamics on organoid morphology at later stages of differentiation, we analysed the inter-line variability at day 90. OS organoids display higher morphological variability between the two lines, particularly in area, perimeter, average radius and curvature-related features, whereas RC organoids exhibit a more uniform profile. Notably, roundness values remain significantly higher in RC organoids across both lines, reinforcing the ability of the RC system to maintain a consistent structural organization (Fig. EV2C). Although some differences between the two hiPSC lines persist, the RC protocol significantly mitigates morphological variability compared to OS, highlighting its robustness in generating reproducible organoids. These findings indicate that the RC methodology effectively standardizes morphometric parameters while also reducing inter-line variability at later differentiation stages. By integrating automated morphological analysis with PCA and direct statistical comparisons, we demonstrate that the RC protocol ensures consistent organoid quality and morphological

reproducibility, establishing a reliable framework for specific applications.

## RC organoids display dorsal forebrain identity

To assess the differentiation and cellular identity in organoids generated using the RC protocol, we harvested them at day 20 (Fig. 4A) and compared them to OS organoids (Fig. 4B–F).

Immunohistological analyses revealed the presence of numerous neuroepithelial rosettes that constituted most of the organoids both in the RC and OS protocol-derived tissues (Fig. 4B). These cellular organizations are a hallmark of early neural patterning and are predominantly positive for PAX6 (Fig. 4B left insets), an established marker of dorsal forebrain progenitors. At the border and outside of the rosettes, we identified newborn neurons positive for the marker DCX (Braun et al, 2023), and negative for the interneuron progenitors marker DLX2, in both protocols (Figs. 4B right insets and EV3A).

To assess the emergence of dorsal patterning, we quantified the percentage of PAX6-positive cells in RC and OS organoids at three different time points—day 25, 45 and 90—to follow the organoids' differentiation (Fig. 4C). We consistently found a higher fraction of PAX6-positive cells in the RC protocol (Fig. 4C right top). To characterize rosettes' morphology and localization, we stained for a tight junction marker ZO1 across the same time points (Figs. 4D and EV3B), which showed their presence both in the center and the periphery of the organoid. While the rosette area was, on average, comparable between RC and OS organoids (Fig. 4D right top), the number of rosettes tended to be higher in the RC protocol across all time points (Fig. 4D right bottom). This suggests a higher level of neurogenesis in the RC protocol. To test whether neural progenitors were more proliferative in the RC protocol, we stained for the proliferation marker Ki67 and quantified the fraction of proliferative cells and of PAX6 progenitors (Fig. 4C right bottom). A larger fraction of cycling progenitors in the RC protocol was found, compared to the OS protocol (Figs. 4C right bottom and EV3C). We then assessed the neurogenic potential of the cortical progenitors in both RC and OS organoids, by quantifying the

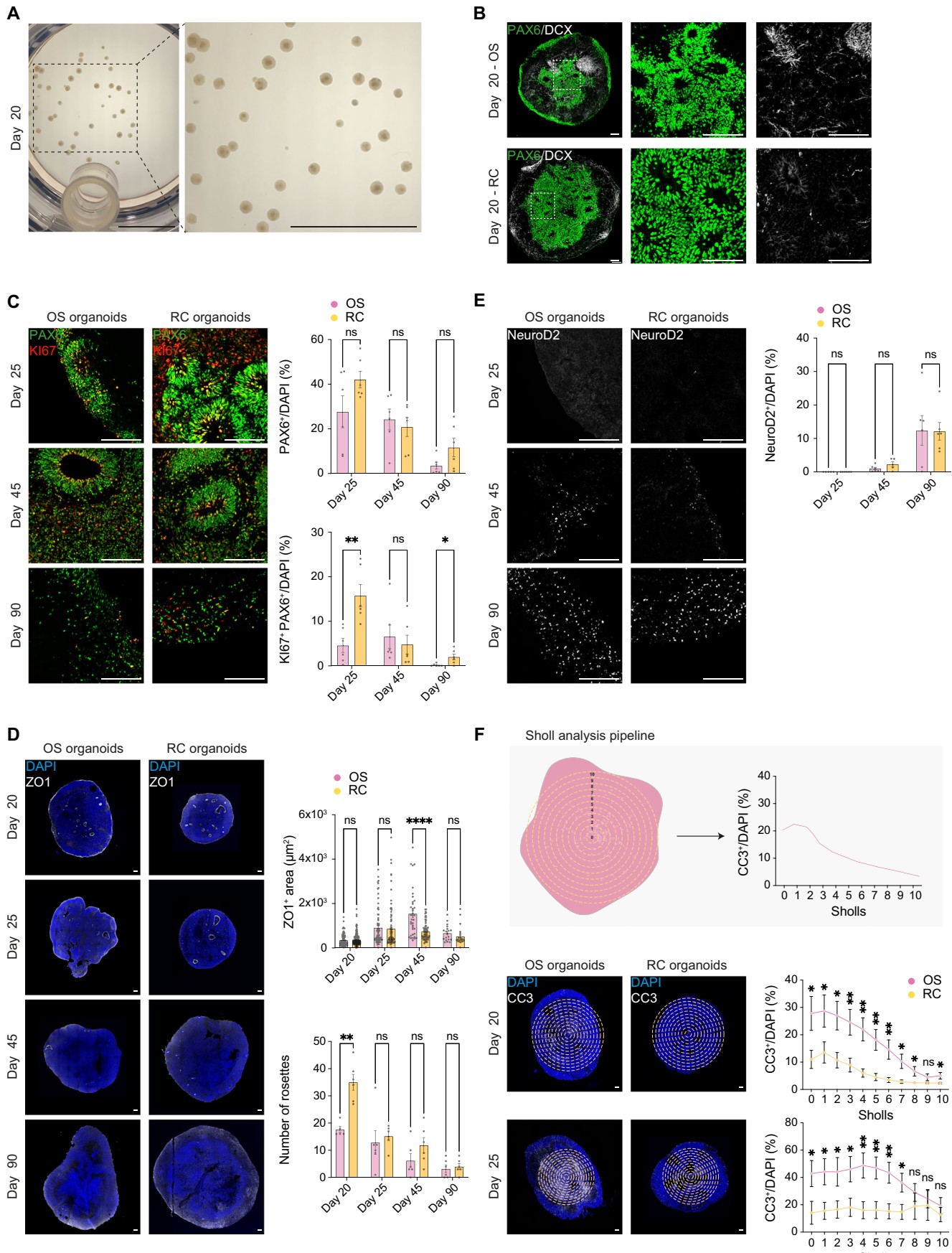

◀

**Figure 4. RC organoids exhibit dorsal forebrain identity.**

(A) Brightfield images of day 20 organoids cultured in the RC system, showing uniform morphology and size distribution. Scale bars: 1 mm. (B) Immunofluorescence images for PAX6 (green) and DCX (white) in OS and RC organoids at day 20. The main panel provides an overview of the organoid structure, while high-magnification insets, highlighted by a dashed-line rectangle, show the distribution of PAX6+ progenitors and DCX+ migrating neurons. Scale bars: 100 μm. (C) Left: Representative images of OS and RC organoids at days 25, 45, and 90 stained for PAX6 (green) and Ki67 (red). Scale bars: 100 μm. Right: Quantification of the proportion of PAX6$^+$/DAPI$^+$ (right top) and Ki67$^+$PAX6$^+$/ DAPI$^+$ (right bottom) cells across time points ($n = 3$ batches per line). Data are presented as mean ± SEM. Statistical analysis was performed using an unpaired two-tailed Welch's $t$-test with correction for multiple comparisons using the two-stage step-up method of Benjamini, Krieger, and Yekutieli. ns not significant, $^*p < 0.05$, $^{**}p < 0.01$. Exact $p$ values: day 25 (PAX6/DAPI$^+$: $p = 0.112862$; Ki67$^+$PAX6$^+$/ DAPI$^+$: $p = 0.004931$), day 45 (PAX6$^+$/DAPI$^+$: $p = 0.602666$; Ki67$^+$PAX6$^+$/ DAPI$^+$: $p = 0.610968$), day 90 (PAX6$^+$/DAPI$^+$: $p = 0.111136$; Ki67$^+$PAX6$^+$/ DAPI$^+$: $p = 0.037715$). (D) Left: Representative images of OS and RC organoids at days 20, 25, 45, and 90 stained for ZO1 (white) and DAPI (blue). Scale bars: 100 μm. Right: Quantification of ZO1$^+$ area (right top) and rosette number (right bottom) across time points. In the top graph, each dot represents the area of a single ZO1$^+$ rosette; in the bottom graph, each dot represents a different batch ($n = 3$ batches per line). Data were presented as mean ± SEM. Statistical analysis was performed using an unpaired two-tailed Welch's $t$-test with correction for multiple comparisons using the two-stage step-up method of Benjamini, Krieger, and Yekutieli. ns not significant, $^{**}p < 0.01$, $^{****}p < 0.0001$. Exact $p$ values: ZO1$^+$ area - day 20 ($p = 0.532809$), day 25 ($p = 0.710857$), day 45 ($p = 0.000065$), day 90 ($p = 0.214874$); Number of rosettes - day 20 ($p = 0.001065$), day 25 ($p = 0.632978$), day 45 ($p = 0.168185$), day 90 ($p = 0.649721$). (E) Left: Representative images of OS and RC organoids at days 25, 45, and 90 stained for NeuroD2 (white). Scale bars: 100 μm. Right: Quantification of the proportion of NeuroD2$^+$/DAPI$^+$ cells across time points ($n = 3$ batches per line). Data were presented as mean ± SEM. Statistical analysis was performed using an unpaired two-tailed Welch's $t$-test with correction for multiple comparisons using the two-stage step-up method of Benjamini, Krieger, and Yekutieli. ns not significant, $^*p < 0.05$, $^{**}p < 0.01$. Exact $p$ values: day 25 ($p = 0.251631$), day 45 ($p = 0.136141$), day 90 ($p = 0.960241$). (F) Top: Schematic representation of the Sholl analysis pipeline used to quantify the spatial distribution of apoptotic cells. Concentric circular shells are drawn starting from the organoid center, with each shell spaced 50 μm apart. The percentage of CC3$^+$/DAPI$^+$ cells is calculated within each shell. Middle: Representative images of OS and RC organoids at day 20 stained for cleaved caspase-3 (CC3, white) and DAPI (blue), with overlay of the Sholl shells. Quantification shows the distribution of CC3$^+$ cells across shells ($n = 3$ batches per line). Scale bars: 100 μm. Bottom: Representative images and quantification of CC3$^+$/DAPI$^+$ cells at day 25. Quantification shows the distribution of CC3$^+$ cells across shells ($n = 3$ batches per line). Scale bars: 100 μm. Data were presented as mean ± SEM. Statistical analysis was performed using an unpaired two-tailed Welch's $t$-test with correction for multiple comparisons using the two-stage step-up method of Benjamini, Krieger, and Yekutieli. ns not significant, $^*p < 0.05$, $^{**}p < 0.01$. Exact $p$ values: day 20 – sholl 0 ($p = 0.024870$), sholl 1 ($p = 0.040448$), sholl 2 ($p = 0.011206$), sholl 3 ($p = 0.007268$), sholl 4 ($p = 0.001842$), sholl 5 ($p = 0.003006$), sholl 6 ($p = 0.006939$), sholl 7 ($p = 0.012676$), sholl 8 ($p = 0.029024$), sholl 9 ($p = 0.120759$), sholl 10 ($p = 0.043613$). Day 25 – sholl 0 ($p = 0.035160$), sholl 1 ($p = 0.021978$), sholl 2 ($p = 0.037193$), sholl 3 ($p = 0.024342$), sholl 4 ($p = 0.008247$), sholl 5 ($p = 0.006828$), sholl 6 ($p = 0.007409$), sholl 7 ($p = 0.024731$), sholl 8 ($p = 0.404360$), sholl 9 ($p = 0.709928$), sholl 10 ($p = 0.423070$). Source data are available online for this figure.

fraction of cells positive for the early cortical neuron marker NeuroD2 (Fig. 4E). We found comparable percentages of neuronal cells in both protocols across all three time points, suggesting that neurogenesis is proportionally conserved in the two protocols, but the higher proliferative state of cortical progenitors in RC-derived organoids generated overall bigger organoids (Fig. EV2B, Area and perimeter measurements).

To assess the health of the RC-derived organoids and the extent of the necrotic core inside the structures compared to the OS protocol, we quantified the cells positive for the cell death marker Cleaved caspase-3 (CC3), with a custom image analysis pipeline (Methods) at days 20 and 25. In brief, organoid sections were stained for CC3 and DAPI, and Sholl analysis (Binley et al, 2014) was used to measure the percentage of CC3$^+$ cells from the center towards the periphery (Fig. 4F top). Our results revealed that the necrotic core is present in both conditions, but there is a significantly reduced amount of CC3$^+$ cells across the whole area analyzed in the RC protocol compared to the OS protocol (Fig. 4F, middle and bottom panels). Overall, these results confirmed that the RC system allows the acquisition of a typical dorsal forebrain organoid morphology and architecture, ensuring uniform tissue organization, and potentially better sustaining a healthy proliferative state and cell viability of cortical progenitors compared to shaker-based methods.

Qualitative analysis across organoid development of other, well-known, markers included SOX2, a marker for multipotent neural stem cells; FOXG1, a transcription factor critical for forebrain specification; SATB2, a marker of upper-layer cortical neurons; and TBR1, a marker of deep-layer cortical neurons; MAP2, a pan-neuronal marker, confirmed dorsal identity (Fig. EV3D). Additionally, ventral forebrain markers, including NKX2.1 (MGE) and DLX2 (interneuron progenitors (Braun et al, 2023)), were not expressed (Fig. EV3D). The absence of ventral markers, combined with the expression of dorsal markers, highlights the robustness of the RC protocol in driving region-specific organoid differentiation.

## RC organoids consistently recapitulate cell types of the developing human brain

To comprehensively characterize the identity of the RC organoids, we evaluated their cellular composition at day 90 using single-cell RNA sequencing technology (Fig. 5A). This time point was chosen as it represents a stage of organoid maturation in which cellular diversity allows for a comprehensive evaluation of neuronal differentiation and progenitor maintenance (Khan et al, 2020; Velasco et al, 2019).

As part of the preprocessing step, we applied the computational algorithm *Gruffi* (Vertesy et al, 2022) (see Methods), which employs granular functional filtering to remove stressed cells in an unbiased manner, thus minimizing confounding variability introduced by stress-specific artifacts (Dony et al, 2025; Kim et al, 2025; Martins-Costa et al, 2023). After filtering, a total of 17'738 high-quality cells were retained for further analysis, providing a refined dataset for downstream comparisons (Fig. EV4A). We annotated the single-cell transcriptomic data from RC organoids using a reference human fetal brain dataset (Polioudakis et al, 2019) (Fig. 5B left and Methods). The fetal dataset includes distinct progenitor populations, including cycling progenitors in both G2/M (PgG2M) and S (PgS) phases, and mixed progenitors (Prog_mixed). Radial glial cells (RGs) include outer radial glia (oRG) and ventricular radial glia (vRG), while excitatory neuronal populations comprise maturing excitatory upper enriched (ExM_U), excitatory deep layer 1 (ExDp1), maturing excitatory (ExM), migrating excitatory (ExN), and

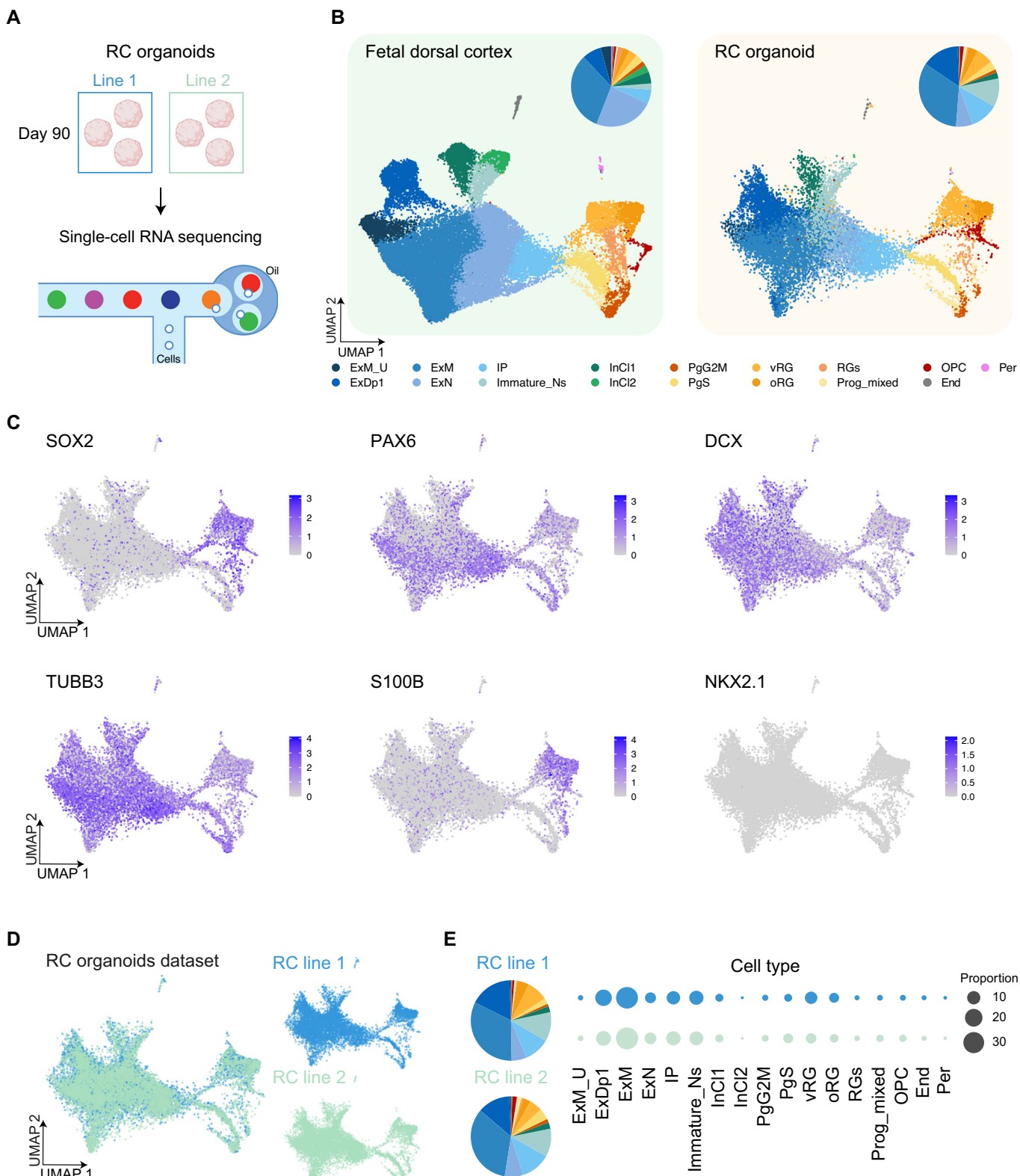

immature neurons (Immature_Ns). In addition, intermediate progenitors (IP), early oligodendrocyte progenitor cells (OPCs), a small population of endothelial (End) cells and pericytes (Per), were identified (Fig. 5B right). We also identified two clusters of

interneurons (Fig. 5B right), showing expression of interneuron markers (Fig. EV4B). Subclustering analysis showed limited overlap between organoid-derived cells and fetal brain-derived interneurons (Fig. EV4C), and a weaker expression of cell type-

**Figure 5. RC organoids recapitulate cortical cell types and exhibit inter-line cellular reproducibility.**

(A) Schematic representation of the experimental workflow. RC organoids from two independent iPSC lines were harvested at day 90, dissociated into single cells, and subjected to single-cell RNA sequencing. (B) UMAP representation of the fetal dorsal cortex (left) (Polioudakis et al, 2019) and RC organoids (right), illustrating the clustering of cell populations. The pie charts indicate the proportional distribution of different cell types in each dataset. (C) Feature plots displaying the expression patterns of key neural progenitor and neuronal markers, including SOX2 (neural progenitors), PAX6 (dorsal progenitors), DCX (migrating neuroblasts), TUBB3 (neuronal marker), S100B (radial glia and oligodendrocyte progenitors), and NKX2.1 (ventral forebrain marker). (D) UMAP representation of the RC organoid dataset, separated by iPSC lines (RC line 1 and RC line 2), demonstrating consistency between replicates. (E) Proportional representation of cell types across the two RC organoid lines, visualized through dot plots and pie charts, confirming the reproducibility of the differentiation process. Cell types include: ExM_U (maturing excitatory upper enriched), ExDp1 (excitatory deep layer 1), ExM (maturing excitatory), ExN (migrating excitatory), IP (intermediate progenitors), Immature_Ns (immature neurons), InCl1 (interneuron cluster 1), InCl2 (interneuron cluster 2), PgG2M (cycling progenitors in G2/M phase), PgS (cycling progenitors in S phase), vRG (ventricular radial glia), oRG (outer radial glia), RGs (radial glial populations), Prog_mixed (mixed progenitors), OPC (oligodendrocyte progenitor cells), End (Endothelial cells), and Per (pericytes).

specific markers (Fig. EV4D), suggesting that these cells are probably immature interneurons (they express GAD1– Fig. EV4D top left), and could be of dorsal origin (Delgado et al, 2022; Kim et al, 2023). Our annotation was supported by similar canonical marker expression between the fetal brain and organoid-derived cells (Fig. EV4B). Neural progenitors exhibited strong expression of SOX2 and PAX6, while neurons displayed robust expression of DCX and TUBB3, confirming the presence of both migrating neuroblasts and maturing neurons (Fig. 5C). Additionally, S100B was expressed in radial glia and OPC. In conclusion, our analysis revealed that RC organoids closely recapitulate all the major cell types of the mid-gestation human cortex (Dataset EV5). To contextualize our analysis and model, we also compared our single-cell data with another published dataset of organoids generated using a similar protocol but without using the RC system -herein referred to as the "KH" dataset (Khan et al, 2020). This dataset was initially preprocessed as the RC organoids dataset and later integrated into the joint human fetal brain and RC organoids dataset for comparison in cellular composition (Fig. EV4E,F). Our analysis showed that the RC organoids-derived cells show comparable diversity to the KH dataset, with slight differences in the relative proportions of some cell types (e.g., ExDp1) across the two cell lines used in the KH dataset (Fig. EV4G). However, sampling bias is one confounding factor in single-cell analyses that should be taken into consideration.

To investigate the functional profiles of the cell types, present in the RC organoid dataset, we identified the most significant marker genes for each cell type (see Methods) and performed Gene Ontology (GO) enrichment analysis for each cell population (Fig. EV5A). The analysis confirmed that the RC organoid cell types are associated with processes relevant to brain development. For example, ExM were enriched for terms such as "axonogenesis" and "regulation of synapse organization". ExM_U and Immature_Ns were enriched for "forebrain development" while progenitor populations, such as oRG, showed enrichment for "regulation of neuron projection development", while processes including "chromosome segregation" were prominent in proliferative populations. These results emphasize the functional alignment of RC organoid cell types with their in vivo counterparts (Polioudakis et al, 2019), further supporting the physiological relevance of RC-generated organoids. We next evaluated the inter-line reproducibility by comparing the proportions of cell types between the two RC lines, using the fetal brain samples as reference (Polioudakis et al, 2019). The UMAP representation demonstrated a strong overlap across all cell types between the two hiPSC lines (Fig. 5D). The proportions of cell types across the two different lines and gestational weeks are

strongly overlapping (Fig. 5E), consistently demonstrating that the cell type distributions were comparable between the two RC lines. Together, these analyses confirm that the RC protocol generates highly reproducible cellular compositions across different genetic backgrounds, with the proportions of key progenitor and neuronal populations being remarkably consistent. Additionally, these findings highlight the inter-line reproducibility of cellular composition in RC organoids.

## Metabolic profiling of RC organoids

Energy metabolism has recently emerged as a key focus area in neuroscience, particularly in the context of neurodevelopmental disorders, where metabolic dysfunctions are increasingly implicated in a wide range of neurological conditions (Camandola and Mattson, 2017; Gandal et al, 2018; Kanellopoulos et al, 2020; Mariano et al, 2023). Moreover, metabolic regulation plays a crucial role in neurodevelopment, influencing processes such as progenitor proliferation, neuronal differentiation, and synaptic activity (Badal et al, 2019; Iwata et al, 2023; Iwata et al, 2020; Khacho and Slack, 2018). However, recent studies have highlighted metabolic heterogeneity in brain organoids, demonstrating how shifting between OXPHOS and glycolysis impacts differentiation outcomes (Øhlenschlæger et al, 2023). Given the transcriptional signatures linked to mitochondrial function and oxidative metabolism observed in RC organoids (Fig. EV5A), we sought to investigate whether specific metabolic pathways, OXPHOS and glycolysis, were differentially regulated across cell types.

The analysis showed that OXPHOS-related genes are broadly expressed across multiple cell types, with particularly high expression in progenitors (PgS and PgG2M) and radial glia (oRG, vRG, and RGS), whereas glycolysis-related genes exhibit a more evenly distributed but lower expression overall (Fig. 6A). This transcriptional evidence indicates that RC organoids predominantly rely on mitochondrial respiration for energy production, although mRNA levels do not always correlate with metabolic activity. Despite the growing interest, measuring oxygen consumption rates (OCR), a critical indicator of mitochondrial activity and metabolic health, is not trivial in brain organoids and, to our knowledge, not well-established. To address this gap, we established a novel protocol using the Oroboros O2K system (Bird et al, 2019) to measure mitochondrial oxygen consumption in the whole RC organoids. We dissociated a pool of three organoids for each of the two lines at day 90 and proceeded with the OCR protocol (see Methods), which captures key metabolic states, including OXPHOS, electron transport system (ETS) capacity, ROX and

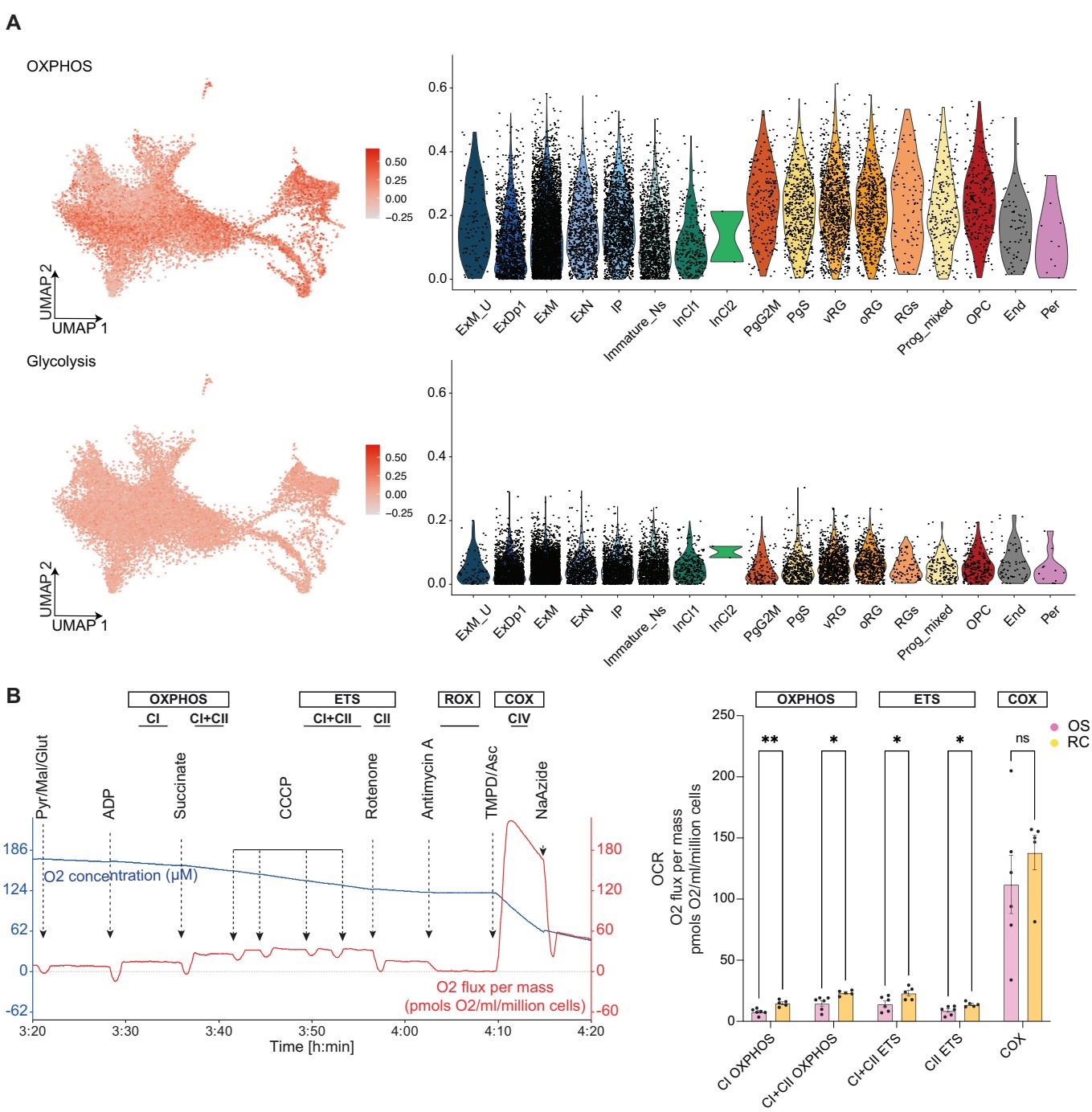

COX activity across distinct mitochondrial complexes (Fig. 6B). Example traces (Fig. 6B, left) illustrate the profile of a well responding pool, providing detailed insight into the functional contributions of complexes I, II, and IV. Quantitative analysis (Fig. 6B, right) revealed that RC organoids exhibit a significantly higher overall OCR compared to OS organoids, indicating a metabolic shift toward enhanced OXPHOS. Notably, OCR levels remained highly consistent across batches and cell lines, further supporting the robustness and reproducibility of mitochondrial function in RC organoids (Fig. 6B, right).

## Discussion

Approximately 90% of clinical candidate drugs fail to progress from preclinical research to market approval (Seyhan, 2019). This high rate highlights a significant gap between preclinical models and real-world patient responses to treatments. Unlike conventional 2D cultures or in vivo models, organoids more accurately mimic human physiology (Antonica et al, 2022; Camp et al, 2015; Eichmuller and Knoblich, 2022; Kelley and Pasca, 2022; Kim et al, 2020; Lancaster et al, 2013; Luo et al, 2016; Pellegrini et al, 2020;

◀ **Figure 6. Metabolic profiling of RC organoids.**

(A) Single-cell transcriptomic analysis of oxidative phosphorylation (OXPHOS) and glycolysis-related gene expression across different cell types in RC organoids (same dataset as Fig. 5). UMAP plots on the left display the expression levels of OXPHOS and glycolysis genes, with intensity represented by a color scale. Violin plots on the right quantify the expression levels across different cell populations, including ExM_U (maturing excitatory upper enriched), ExDp1 (excitatory deep layer 1), ExM (maturing excitatory), ExN (migrating excitatory), IP (intermediate progenitors), Immature_Ns (immature neurons), InCl1 (interneuron cluster 1), InCl2 (interneuron cluster 2), PgG2M (cycling progenitors in G2/M phase), PgS (cycling progenitors in S phase), vRG (ventricular radial glia), oRG (outer radial glia), RGs (radial glial populations), Prog_mixed (mixed progenitors), OPC (oligodendrocyte progenitor cells), End (Endothelial cells) and Per (pericytes). (B) Oxygen consumption rate (OCR) measurements in RC organoids at day 90 using high-resolution respirometry with the Oroboros O2K system. The panel on the left displays representative OCR traces, illustrating different mitochondrial respiration states: complex I oxidative phosphorylation (CI OXPHOS), combined complex I and complex II oxidative phosphorylation (CI + CII OXPHOS), combined complex I and complex II electron transport system (CI + CII ETS), complex II electron transport system (CII ETS), and cytochrome c oxidase activity (COX). Sequential compound additions, including ADP, succinate, CCCP, rotenone, antimycin A, and sodium azide, allow for the functional dissection of mitochondrial respiratory chain components. The panel on the right presents a quantitative comparison of OCR levels between RC organoids (yellow) and OS organoids (pink), normalized to cell number. Each dot represents a pool of three organoids from a different batch. Statistical analysis was performed using multiple unpaired $t$-tests with Welch's correction. Error bars represent the standard error of the mean (SEM). ns not significant, $*p < 0.05$, $**p < 0.01$, $***p < 0.001$, $****p < 0.0001$. Exact $p$ values: CI OXPHOS ($p = 0.001581$), CI + CII OXPHOS ($p = 0.011799$), CI + CII ETS ($p = 0.029318$), CII ETS ($p = 0.019172$), COX ($p = 0.374416$). Source data are available online for this figure.

Qian et al, 2019). However, the reproducibility of brain organoids remains a fundamental challenge, as each single organoid has a different overall cytoarchitecture and morphology, making it a major determinant of experimental variability.

Early tissue architecture plays a decisive role in guiding organoid development. Notably, structured aggregation methods, such as those employing microwell platforms (e.g., AggreWell), have improved control over initial organoid size and cellular composition, thereby minimizing heterogeneity (Gjorevski et al, 2022). Our results highlight that an initial extended 6-day aggregation period significantly favors organoid shape regularity and reproducibility, which could be beneficial for certain specific applications. However, it does not automatically mean that the organoid tissue is of high-quality, and in-depth characterization is always required.

Organoid cytoarchitecture is linked to developmental health, but also influences downstream differentiation, regional patterning and functional properties (Chiaradia et al, 2023; Jain et al, 2025; Lancaster et al, 2017; Martins-Costa et al, 2023). For this reason, significant attention has been given to the biomechanical forces that act during organoid formation. The fFSS, a biomechanical force generated by media flow in dynamic culture systems, has been recognized as a key factor influencing morphogenesis (Dahl-Jensen and Grapin-Botton, 2017; Goto-Silva et al, 2019; Hinton et al, 2022). While dynamic systems, such as orbital shakers and bioreactors, enhance nutrient and oxygen delivery, they also expose developing organoids to potentially disruptive fFSS (Dardik et al, 2005; Gareau et al, 2014; Ismadi et al, 2014; Saglam-Metiner et al, 2023; Suong et al, 2021). To mitigate the effects of fFSS, optimization of spinning speed in bioreactors, usage of modified orbital shakers with reduced turbulence, or of static culture systems that eliminate shear forces entirely were tested. However, these approaches often come with drawbacks, such as uneven nutrient distribution in static cultures or residual variability in dynamic systems (Dardik et al, 2005; Gareau et al, 2014; Ismadi et al, 2014; Saglam-Metiner et al, 2023; Suong et al, 2021). Our study builds on this foundation by implementing the RC system, which drastically reduces fFSS thanks to gentle rotational dynamics, promoting optimal tridimensional organoid development, ultimately better balancing nutrient delivery and mechanical stability. While we demonstrated that the morphology of the RC-generated organoids is consistent throughout development, future investigation could assess more in-depth the molecular differences

between RC and OS-derived organoids. Moreover, using the RC system, we achieved high inter-batch and inter-line reproducibility, as evidenced by consistent morphological parameters, while obtaining a cellular composition representative of fetal human brain cell type diversity at mid-gestation, similar to other previously used organoid protocols, and improved overall cell survival.

This level of consistency enhances the utility of RC organoids for applications requiring reliable and reproducible models, such as high-throughput drug screening and disease modeling. In addition, these findings emphasize the interplay between forces and morphology in organoid development and suggest that controlling biomechanical conditions can significantly improve reproducibility.

RC-derived organoids were found to be metabolically active, as with our novel OCR protocol, we could measure robust mitochondrial function with low variability, across organoids derived from different cell lines. Importantly, RC organoids predominantly rely on OXPHOS rather than glycolysis, different from organoids obtained using the standard guided protocol (Øhlenschlæger et al, 2023), and consistent with brain development, where neuronal differentiation is accompanied by a metabolic shift from glycolysis toward OXPHOS (Agostini et al, 2016). This further corroborates the notion that biomechanical forces during key morphogenesis phases might have an impact on metabolic states, and consequently, tissue fate. Future experiments should dissect in detail the metabolic heterogeneity across different neural populations during human brain development and compare this to organoid development. However, our current dataset does not allow us to conclude whether the observed increase in OXPHOS reflects enhanced neuronal maturation, more closely resembling the physiological state, or simply improved overall organoid health.

Our study highlights the advantages of the RC system for dorsal forebrain organoid generation; however, there are limitations that could be addressed with future investigations. To balance reproducibility and practical feasibility, we transitioned organoids from the RC system to an orbital shaker after day 25. This choice was driven by both biological and logistical considerations. Over extended culture periods, organoids increase significantly in size, reaching up to 1 cm, thereby raising the risk of physical contact and fusion events within the confined dimensions of the RC chamber. Additionally, the RC platform is designed for optimal performance within a two-week window, necessitating regular replacement. A long-term organoid culture would pose substantial cost and handling challenges, but future research could explore whether prolonged

RC-based culture could offer additional advantages, not only in terms of reproducibility, but also regarding improved patterning and maturation, to more closely mimic human brain development.

Emerging studies suggest that fine-tuning the molecular and environmental cues during differentiation is crucial for achieving more accurate regional identities (Rosebrock et al, 2022). Incorporating molecular adjustments into our protocol could further refine the developmental trajectories of organoids, ensuring more faithful recapitulation of spatiotemporal patterns and cell fate specification observed in the human brain.

Another important aspect is the use of ECM components such as Matrigel, and their impact on organoid morphology and differentiation outcomes (Pagliaro et al, 2025). While Matrigel supports early tissue structuring, its compositional differences from brain-specific ECM may affect endogenous ECM production, influencing developmental fidelity and thus contributing to variability in differentiation outcomes (Jain et al, 2025; Long et al, 2018; Martins-Costa et al, 2023). Current research focuses on developing more standardized and compositionally defined ECM scaffolds that mimic native tissue-specific matrices (Pagliaro et al, 2025) that could be tested in combination with the RC system.

In conclusion, by addressing the challenges of morphological variability, we have provided a framework for brain organoid generation, positioning the use of the RC system as a valuable alternative for studying neurodevelopment, modeling diseases, and testing therapeutic interventions.

## Methods

### Reagents and tools table

| Reagent/resource | Reference or source | Identifier or catalog number |
| --- | --- | --- |
| **Media and reagents (cell culture)** | | |
| Essential 6™ Medium Kit | Thermo Fisher Scientific | Cat# A1516401 |
| Essential 8™ Medium Kit | Thermo Fisher Scientific | Cat# A1517001 |
| Neurobasal™ -A Medium | Thermo Fisher Scientific | Cat# 10888-022 |
| B-27™ Plus Supplement (50x) | Thermo Fisher Scientific | Cat# A3582801 |
| B-27™ Supplement minus vitamin A (50x) | Thermo Fisher Scientific | Cat# 12587010 |
| cAMP (N6,2'-O-dibutyryl adenosine-3',5' (cyclic)-monophosphate sodium salt) | Merck | Cat# D0627-25MG |
| DHA (cis-4,7,10,13,16,19-Docosahexaenoic acid) | Merck | Cat# D2534-25MG |
| DSM (dorsomorphin) | Merck | Cat# P5499-5MG |
| GlutaMAX™ Supplement 100X | Thermo Fisher Scientific | Cat# 35050038 |
| Human/Mouse/Rat BDNF Recombinant Protein, ReproTech® | Thermo Fisher Scientific | Cat# 45002 |

| Reagent/resource | Reference or source | Identifier or catalog number |
| --- | --- | --- |
| Human EGF, Animal-free recombinant protein, ReproTech® | Thermo Fisher Scientific | Cat# AF10015 |
| Human FGF-basic (FGF2/bFGF) (154 amino acid) Recombinant protein, ReproTech® | Thermo Fisher Scientific | Cat# 10018B |
| Human NT-3 Recombinant protein, ReproTech® | Thermo Fisher Scientific | Cat# 45003 |
| 2-phospho-L-ascorbic acid trisodium salt (L-AA) | Merck | Cat# 49752 |
| Penicillin-Streptomycin (10,000 U/ml) | Thermo Fisher Scientific | Cat# 15140122 |
| SB-431542 | R&D Systems | Cat# 1614 |
| Y-27632 | Tebu-Bio | Cat# S1049 |
| Accutase | StemCell technologies | Cat# 7922 |
| Anti-adherence rinsing solution | StemCell technologies | Cat# 07010 |
| DMEM/F12, HEPES (1x) | Thermo Fisher Scientific | Cat# 11330032 |
| DPBS, calcium, magnesium (1x) | Thermo Fisher Scientific | Cat# 14040091 |
| DPBS, no calcium, no magnesium (1x) | Thermo Fisher Scientific | Cat# 14190144 |
| Geltrex™ LDEV-Free, hESC-qualifies, Reduced growth factor basement membrane matrix | Thermo Fisher Scientific | Cat# A1413301 |
| ReLeSR™ | StemCell technologies | Cat# 100-0483 |
| Trypsin, EDTA (0.25%), phenol red | Thermo Fisher Scientific | Cat# 25200056 |
| AggreWell™ 800 | StemCell technologies | Cat# 34815 |
| Clinoreactor (CelVivo plate) | CelVivo | Cat# 30003 |
| Ethanol | Merck | Cat# 51976 |
| Cellstar®, cell-repellent surface, 6 cm | Greiner Bio-One GmBH | Cat# 628979 |
| Cellstar®, cell-repellent surface, 10 cm | Greiner Bio-One GmBH | Cat# 664970 |
| **Antibodies** | | |
| PAX6 antibody (1/500) | Biolegend | Cat# 901301 |
| PAX6 antibody (1/200) | BD pharmingen | Cat# 561462 |
| FOXG1 antibody (1/200) | Abcam | Cat# ab18259 |
| SOX2 antibody (1/200) | Abcam | Cat# ab171380 |
| Doublecortin (E-6) antibody (1/100) | Santa Cruz | Cat# sc-271390 |
| TBR1 antibody (1/200) | Abcam | Cat# ab31940 |
| SATB2 antibody (1/100) | Invitrogen | Cat# MA5-32788 |
| NeuroD2 antibody (1/500) | Abcam | Cat# ab104430 |

| Reagent/resource | Reference or source | Identifier or catalog number |
|---|---|---|
| MAP2 antibody (1/500) | Sigma | Cat# M4403 |
| NKX2.1/TTF-1 antibody (1/200) | Abcam | Cat# ab133737 |
| DLX2 antibody (1/200) | Santa Cruz | Cat# sc-393879 |
| Cleaved caspase-3 antibody (1/400) | Cell Signalling | Cat# 9661 |
| ZO1 antibody (1/200) | Thermo Fisher | Cat# 61-7300 |
| **Secondary antibodies** | | |
| Goat anti-mouse IgG (H + L), Alexa Fluor™ 488 | Invitrogen | Cat# A-11029 |
| Goat anti-mouse IgG (H + L), Alexa Fluor™ 546 | Invitrogen | Cat# A-11030 |
| Goat anti-mouse IgG (H + L), Alexa Fluor™ 647 | Invitrogen | Cat# A-21236 |
| Goat anti-rabbit IgG (H + L), Alexa Fluor™ 488 | Invitrogen | Cat# A-11034 |
| Goat anti-rabbit IgG (H + L), Alexa Fluor™ 546 | Invitrogen | Cat# A-11035 |
| Goat anti-rabbit IgG (H + L), Alexa Fluor™ 647 | Invitrogen | Cat# A-21244 |
| **Single-cell RNA sequencing** | | |
| Chromium Next GEM Chip G Single Cell Kit | 10x Genomics | Cat# PN-1000127 |
| Chromium Next GEM Single Cell 3′ Kit v3.1 | 10x Genomics | Cat# PN-1000269 |
| **Oroboros O2K reagents** | | |
| MiR05 buffer | Oroboros Instruments | Cat# 60101-01 |
| Digitonin | Sigma-Aldrich | Cat# D5628 |
| Malate | Sigma-Aldrich | Cat# M1000 |
| Pyruvate | Sigma-Aldrich | Cat# P2256 |
| Glutamate | Sigma-Aldrich | Cat# G1626 |
| ADP | Calbiochem | Cat# 117105 |
| Succinate | Sigma-Aldrich | Cat# S2378 |
| Carbonyl cyanide m-chlorophenyl hydrazone (CCCP) | Sigma-Aldrich | Cat# C2759 |
| Rotenone | Sigma-Aldrich | Cat# R8875 |
| Antimycin A | Sigma-Aldrich | Cat# A8674 |
| Sodium ascorbate | Sigma-Aldrich | Cat# A4034 |
| *N,N,N′,N′*-tetramethyl-p-phenylenediamine dihydrochloride (TMPD) | Sigma-Aldrich | Cat# T3134 |
| Sodium azide | Sigma-Aldrich | Cat# S8032 |
| **Products** | | |
| Base Mould Disposable 15 × 15 × 06 mm | Kaltek | Cat# 2781 |

| Reagent/resource | Reference or source | Identifier or catalog number |
|---|---|---|
| DAPI | Thermo Fisher Scientific | Cat# 62247 |
| D-sucrose | Roth | Cat# 4621.1 |
| DPBS | Gibco | Cat# 14190-094 |
| Goat serum | Sigma-Aldrich | Cat# G9023-102 |
| PFA 16% | Electron Microscopy Sciences | Cat# 15710-S |
| SDS | Sigma-Aldrich | Cat# L3771-500G |
| Skin porcine gelatine | Sigma-Aldrich | Cat# 48722-500G-F |
| Mowiol Mounting Medium | Homemade | - |
| Triton 10x | Sigma-Aldrich | Cat# T9284-100ML |
| GentleMACS 25 C Tubes | Miltenyi Biotec | Cat# 130-093-237 |
| DNase I, RNase-free (1 U/µL) | Thermo Fisher Scientific | Cat# EN0521 |
| **Instruments** | | |
| Clinostar 2 bioreactor | CelVivo | |
| Echo Revolve2-K2-1861 microscope | Bico Groups | |
| Leica SP8 confocal microscope | Leica Microsystems | |
| **Software/ Python script** | | |
| Biorender | https://app.biorender.com/ | |
| Necrotic core quantification script | https://github.com/ariannaravera/organoids-characterization | |
| Organoid morphology script | https://github.com/ariannaravera/organoids-characterization | |
| Lumen ZO1 quantification script | https://github.com/ariannaravera/organoids-characterization | |
| Histological quantification script | https://github.com/ariannaravera/organoids-characterization | |

## Human induced pluripotent stem cell (hiPSCs) culture

The human iPSC lines were kindly provided by Prof. Peng Jin (Emory University) and previously generated from skin biopsy samples of age-matched healthy males at Emory University (Kang et al, 2021). All iPSC lines were cultured in feeder-free conditions on Geltrex (Gibco)-coated cell culture dishes in E8 Basal Medium (Thermo Fisher Scientific) with 100 U/ml penicillin (Thermo Fisher Scientific) and 100 µg/ml streptomycin (Thermo Fisher Scientific), at 37 °C in 5% CO$_2$. All human pluripotent stem cell lines were maintained below passage 50 and were negative for mycoplasma.

## Brain organoid generation and differentiation using the RC system (RC protocol)

The organoid generation protocol was adapted and modified for RC-harvested organoids from Sloan et al, 2018 (Sloan et al, 2018) as follows. hIPSC at 70-80% of confluency were washed in DPBS-/- (Thermo Fisher Scientific) and incubated with Accutase (StemCell Technologies) at 37 °C for 5 min in a 5% $CO_2$ incubator until the cells detached with gentle shaking. The cell suspension was transferred to a 15 mL conical tube containing an equal volume of DMEM/F12 (Thermo Fisher Scientific) to neutralize the enzymatic activity. Cells were centrifuged at 200×$g$ for 5 min, resuspended in the required volume of E8 medium (Thermo Fisher Scientific) supplemented with Rock Inhibitor (Y-27632) (Tebu-Bio), and adjusted to a final density of 2.5–3 million cells per mL. For EB formation, 1 mL of the single-cell suspension was transferred into a well of an AggreWell plate (StemCell Technologies), achieving a final volume of 1.5 mL per well. EBs were visible within 24 h after seeding and were maintained in AggreWell plates until day 6. On day 1 post-seeding, half of the medium was replaced with Essential 6 medium supplemented with 2.5 μM Dorsomorphin (Merck) and 10 μM SB-431542 (R&D Systems). Half of the medium was changed daily until day 6, except for day 3. At day 6, EBs were transferred from the AggreWell plates to RC Rotating Chambers (CelVIVO) (previously prepared, as described below) containing 15 mL of Neurobasal A-Minus medium. This medium was composed of Neurobasal A medium (Thermo Fisher Scientific) supplemented with B-27 without vitamin A (2%) (Thermo Fisher Scientific), Glutamax (1%) (Thermo Fisher Scientific), EGF (20 ng/mL) (Thermo Fisher Scientific), FGF2 (20 ng/mL) (Thermo Fisher Scientific), and P/S. The medium was changed every 2 days and was maintained until day 25. From day 25 to day 45, organoids were transferred to orbital shakers in 6- or 10-cm dishes and cultured in Neurobasal-A medium supplemented with B-27 without vitamin A (2%) (Thermo Fisher Scientific), Glutamax (1%), BDNF (20 ng/mL) (Thermo Fisher Scientific), NT-3 (20 ng/mL) (Thermo Fisher Scientific), L-ascorbic acid (200 μM) (Merck), DHA (10 μM) (Merck), and cAMP (50 μM) (Merck). The medium was changed three times per week. Starting from day 45, organoids were maintained in Neurobasal A-Plus medium supplemented with B-27 Plus (2%) and Glutamax (1%), without additional growth factors. The medium was changed twice a week to support long-term organoid survival and maturation.

## RC preparation

One day before transferring the EBs, RC systems (CelVIVO) blue beads were hydrated with 25 mL of sterile ultra-pure water, and the chambers were exposed to UV light overnight for sterilization. On the following day, the chambers were thoroughly washed with DPBS (−/−) and DMEM/F12 before adding 10 mL of Neurobasal A-Minus medium. The EBs were then carefully transferred into the prepared chambers and placed in the ClinoStar bioreactor for culture.

## PA protocol

EBs were generated after a 1-day aggregation period in AggreWell plates followed by transfer to low-attachment 10 cm plates, as described in Khan et al, 2020; Sloan et al, 2018. EBs were then cultured until day 6.

## OS protocol

EBs underwent a 6-day aggregation period in AggreWell plates and, starting from day 6, they were cultured in low-attachment 10 cm plates on an orbital shaker using the same media formulation at each corresponding time point as in the RC protocol.

## Computational fluid dynamics (CFD) simulation

CFD simulations were performed using a Navier-Stokes-based solver to model fluid velocity and shear stress for constructs with different density related to compactness of the construct affecting relative weight/buoyancy of constructs with the same surface area (relative mass 1010, 1020, and 1030 where 1000 is density/relative mass of surrounding medium, generated in COMSOL Multiphysics 6.2 by Resolvent Denmark PS, Maaloev, Denmark). Shear force distribution was computed based on the chamber's geometry, while the velocity profiles were extracted to evaluate the stability of the system. Data were plotted to compare velocity and shear stress from the drag force across different conditions.

## Immunohistochemistry

Organoids were fixed in 4% paraformaldehyde for 1 h at 4 °C. After fixation, organoids were washed two times with PBS for 10 min each at room temperature and then incubated in 30% sucrose at 4 °C until they fully sank. Organoids were embedded in a 7.5% gelatin (Sigma-Aldrich) solution containing 10% sucrose (Sigma-Aldrich) and sectioned at 20 μm using a Cryostat. For immunohistochemistry, sections were blocked and permeabilized in a solution of 0.3% Triton X-100 (Sigma-Aldrich) with 3% goat serum (Sigma-Aldrich) for 1 h at room temperature. Primary antibodies were diluted in 0.3% Triton X-100 (Sigma-Aldrich) with 3% goat serum (Sigma-Aldrich) in PBS and incubated overnight at 4 °C. Following primary antibody incubation, sections were washed three times in PBS. Secondary antibodies conjugated with Alexa Fluor 488, 568, or 647 (Invitrogen) were applied for 1 h at room temperature, followed by three additional PBS washes. DAPI (Thermo Fisher Scientific) was included in the secondary antibody solution to stain nuclei. Finally, slides were mounted using Mowiol mounting medium.

The following antibodies were used to perform immunostainings: PAX6 antibody (901301; Biolegend; 1/500); PAX6 antibody (561462; BD pharmingen;1/200); FOXG1 antibody (ab18259; Abcam; 1/200); SOX2 antibody (ab171380; Abcam; 1/200); Doublecortin (E-6) antibody (sc-271390; Santa Cruz; 1/100); TBR1 antibody (ab31940; abcam; 1/200); SATB2 antibody (MA5-32788; Invitrogen; 1:100); NeuroD2 antibody (ab104430; abcam;1/500); MAP2 antibody (M4403; Sigma; 1/500); NKX2.1/TTF-1 antibody (ab133737; Abcam; 1/200); DLX2 antibody (sc-393879; Santa Cruz; 1/200); Cleaved caspase-3 antibody (9661; Cell Signalling;1/400); ZO1 antibody (61-7300; Thermo Fisher; 1/200).

## Imaging

Images were acquired with Echo Revolve2-K2-1861 and Leica SP8 confocal microscopes and processed with Fiji ImageJ.

## Analysis of morphological parameters

Organoids were imaged at day 6 and day 90 using the Echo Revolve2-K2-1861 microscope (Bico Group) in brightfield mode with a 2x objective. The images were converted to TIFF format and analysed using a Python-based script. The organoid outlines, representing the Region of Interest (ROI), were automatically detected by the script and saved into new images for further verification. If the automatically detected contour did not perfectly match the organoid boundary, the threshold was adjusted within the script. In cases where a correct ROI could not be obtained, the organoid with abnormal segmentation was removed from the dataset by verifying its ID number in the result images. Across all conditions, the number of batches varied from three to five per time point and protocol, with a minimum of five organoids analysed per batch.

## Analysis related to morphological parameters

Ten morphological parameters, first described by Chiaradia et al (Chiaradia et al, 2023), were measured from brightfield images of brain organoids and are defined as the following:

Area ($\mu m^2$): Area of the segmented organoid; Perimeter ($\mu m$): Perimeter of the segmented organoid; average radius R0 ($\mu m$): Average distance of the contour from the center of the segmented organoid; Roundness (%): Calculated as $4\pi\times$Area/Major axis$^2$. A roundness of 100% represents a perfect circle; Minimum Feret ($\mu m$): The minimum distance between any two points along the contour; Maximum Feret ($\mu m$): The maximum distance between any two points along the contour; Mean curvature ($\mu m^{-1}$): Average of the curvature along the contour. The curvature is calculated as the inverse of the radius of the osculating circle and is computed as $[(dx \times dyy) - (dy \times dxx)]/[(dx^2) + (dy^2)]^{3/2}$, where dx and dy are the first derivatives in x and y, and dxx and dyy are the second derivatives of the contour; Standard curvature ($\mu m^{-1}$): Standard deviation of the curvature along the contour; Standard curvature $\times$ R0: Standard deviation of the curvature along the contour normalized by the average radius (R0); Dirichlet normal energy (DNE): quantitative descriptor of local curvature variations along the organoid boundary, reflecting the sharpness or complexity of surface contours. DNE is computed as the logarithm of the square of the variation of the normal $n = (dy, -dx)$ of the contour projected on its tangent $t = (dx, dy)$, where dx and dy are the first derivatives in x and y. As the values did not follow a normal distribution and contained some extreme values (notably in PA and OS organoids, because of their inner heterogeneity), Mann–Whitney test (wilcox.test) was applied to compare the morphological parameters between two protocols. The measured values were represented with bar plots. The variability between each protocol was analysed with the Brown-Forsythe test, using the LeveneTest function (center = median). The inter-batch variability was compared with Mann–Whitney test (wilcox.test) and was illustrated in bar plots with the median absolute deviation (MAD), more precisely the absolute value of the MAD divided by the median (Dumitrascu et al, 2019). PCA graphs were performed using the PCA and fviz_par functions. To assess the importance of the contribution of each parameter according to the first three dimensions of the PCA plots, correlation plots were generated using the corrplot function on the cosine2 value (calculated with the get_pca_var function).

## Quantification of immunofluorescent markers

To quantify marker-positive cells in defined regions of the organoids, a semi-automated analysis pipeline was developed in Python. Organoids were stained to detect Ki67, NeuroD2, PAX6 proteins, and DAPI, and imaged using a Leica SP8 confocal microscope. For analysis, images were exported as TIFF files. For each image, a first step involved estimating the average nuclear size (DAPI staining) based on manually annotated regions. This value was later used to infer cell numbers from the total labeled area. Each fluorescence channel was preprocessed individually: the DAPI channel was binarized via Otsu thresholding following Gaussian filtering, while the Ki67, NeuroD2, and PAX6 channels were segmented manually using an interactive GUI implemented in napari, allowing to apply thresholds and review the masks. All segmentation masks were saved and stored. Subsequently, ROIs were manually drawn on the DAPI image to delineate specific domains within each organoid. Within each ROI, the number of positive cells per channel was estimated by dividing the labeled area by the previously computed average nuclear size. In addition to individual marker quantification, colocalization analysis was performed between Ki67 and PAX6 channels to identify proliferative progenitors. The overlapping signal was quantified using binary mask intersection within each ROI.

## Quantification of ZO1+ rosettes

To quantify the number and organization of ZO1$^+$ apical rosettes, a semi-automated image analysis pipeline was implemented in Python. Organoids at days 20, 25, 45, and 90 were stained for ZO1 and DAPI and imaged using a Leica SP8 confocal microscope. For the analysis, images were exported as TIFF files. Images were preprocessed by extracting the ZO1 channel and applying intensity scaling followed by Gaussian filtering. A binary mask of ZO1$^+$ regions was then generated via adaptive thresholding (Otsu method) and refined using morphological operations to remove small objects and fill holes. This preliminary mask was used as input for a semi-manual correction step in napari, where ZO1 segmentation was reviewed and refined. After correction, each image was analyzed to extract: (i) the total number of ZO1$^+$ rosettes, (ii) the area of each rosette, and (iii) its distance from the organoid center of mass (computed from the DAPI mask). All results were exported to CSV format for downstream statistical analysis.

## Sholl analysis pipeline

To assess the spatial distribution of apoptotic cells within organoids, a custom Sholl-based analysis was implemented using a Python script. Organoids at day 20 and day 25 were stained for cleaved caspase-3 (CC3) and DAPI and imaged using a Leica SP8 confocal microscope. For analysis, images were exported as TIFF files. For each image, the organoid boundary was detected from the DAPI channel using adaptive Gaussian thresholding. The center of mass of the organoid was automatically computed from the binary mask, and concentric circles were drawn outward from this center, with a fixed radial increment of 50 $\mu m$. Each ring thus represented a radial shell corresponding to a specific distance from the organoid center. To quantify apoptotic cells, the CC3-positive signal was segmented by

Gaussian filtering, followed by Otsu thresholding and labeling of connected components. Only objects above a defined size threshold were considered. For each ring, the number of CC3⁺ elements was counted, and in the case of larger connected objects, a correction factor was applied to estimate the equivalent number of cells based on area. The number of CC3⁺ elements in each concentric ring was then normalized and exported for statistical analysis. Images with failed segmentation or off-center ROIs were excluded after visual validation.

## Single-cell RNA sequencing

Organoids were cultured in Neurobasal A-Plus medium supplemented with B-27 Plus (2%) and Glutamax (1%) until harvesting at day 90. Pool of three organoids were separately incubated in Trypsin (Sigma-Aldrich)/Accutase (StemCell Technologies) (1:1) containing 10 U/ml DNase I (Thermo Fisher Scientific) in the gentleMACS™ Dissociator (Miltenyi Biotec) set at program NTDK1. After digestion, the cell suspension was passed through a 70-μm strainer. Samples were loaded to recover 16,000 cells onto a Chromium Next GEM Chip G Single Cell Kit (10x Genomics, PN-1000127) and processed through the Chromium controller to generate single-cell GEMs (gel beads in emulsion). scRNA-seq libraries were prepared with the Chromium Next GEM Single Cell 3' Kit v3.1 (10x Genomics, PN-1000269). Organoids libraries were pooled and sequenced using the Element Biosciences AVITI. A total of 166 million reads were requested per library to ensure sufficient coverage for downstream analysis.

## Single-cell RNA sequencing analysis

We aligned reads to the GRCh38 human reference genome with Cell Ranger 7.2 (10x Genomics) using default parameters to produce the cell-by-gene, Unique Molecular Identifier (UMI) count matrix.

UMI counts were analysed using the Seurat R package v.5. Cells were filtered for a min. 1000 genes, maximal mitochondrial content of 10%. Resulting high-quality cells were normalized ("LogNormalize") for scaled for each cell to a total expression of 10 K UMI. Doublets were removed using DoubletFinder, while unlabeled negative cells were retained as their distribution was similar to that of singlets. All analyses were conducted using doublet-free datasets.

## Stress cell identification and removal using *Gruffi*

Brain organoids, unlike fetal brain tissue, frequently exhibit a distinct cellular stress signature, which can interfere with the accurate interpretation of lineage trajectories and developmental fidelity. To ensure data quality and remove stress-affected cells, we applied *Gruffi* algorithm (Vertesy et al, 2022). Stress-affected cells accounted for 15.1% of the dataset and were removed before further analysis. Following *Gruffi* filtering, the cleaned dataset contained 17,738 high-quality cells, ensuring that downstream analyses were performed on physiologically relevant populations.

## Integration with fetal brain and cell type annotation

To validate the identity of organoid-derived cell populations, we integrated our dataset with single-cell RNA-seq data from human fetal cortex at gestational weeks 17–18 (Polioudakis et al, 2019). In brief, the dataset was integrated using the Canonical Correlation Analysis (CCA) based integration within the Seurat package in RStudio and cluster-based annotation was transferred from the fetal data to the organoid data. Previous comparative transcriptomic analyses have shown that day 90 cortical organoids recapitulate the transcriptional landscape of the fetal brain at mid-gestation (Tanaka et al, 2020). This allowed for a direct comparison between organoid and fetal cortical development, ensuring accurate cell type annotation.

## Visualization and analysis

UMAP projections were computed following CCA-based integration of organoid-derived single-cell RNA-seq data with human fetal brain datasets. Cluster identities were assigned using Seurat's RenameIdents() function, and cell types were factorized to ensure consistent ordering across datasets. Feature plots were generated to visualize the expression of key marker genes, including SOX2, PAX6, NKX2-1, DCX, TUBB3, and S100B, across different cell types. Expression levels were mapped onto UMAP projections using Seurat's FeaturePlot() function. To quantify cell-type proportions across batches and conditions, metadata was extracted and processed using group_by() and summarize() functions in tidyverse. Bar plots and stacked bar plots were created to visualize the relative frequencies of each cell type in different datasets. Additionally, pie charts were generated by plotting cell-type proportions in polar coordinates using ggplot2. Dot plots were used to compare cell-type proportions across experimental conditions, where dot size represents the percentage of cells in each category. The dataset was transformed using melt() and plotted using ggplot2. This combination of visualizations provided a detailed characterization of cell-type heterogeneity, lineage specification, and transcriptional states across organoid samples.

## Interneuron subclustering

To refine the interneuron identity, cells annotated as InMGE, InCGE, and Immature_Ns were subsetted and analyzed independently. Layered assay data were normalized using NormalizeData(), followed by variable gene selection (FindVariableFeatures()), scaling (ScaleData()), and dimensionality reduction via PCA. Clustering was performed with FindNeighbors() and FindClusters() using the first 15 PCA components and visualized with UMAP (RunUMAP()). Datasets were then integrated via CCA (IntegrateLayers()) using the integrated.cca reduction. Subclusters were visualized and compared across datasets using DimPlot() and FeaturePlot(). Gene expression (e.g., LHX6, SOX6, NKX2-1, and PROX1) was further summarized via DotPlot().

## Dataset integration and comparative analysis

Raw count matrices from the KH organoid dataset (Khan et al, 2020) were imported using Read10X() and processed in Seurat. Following quality control filtering, datasets were merged, normalized, and dimensionally reduced as described above. Stress-associated transcriptional signatures were removed using the *Gruffi* package (Vertesy et al, 2022). The KH dataset was then annotated using integration with the GE dataset via CCA and mapped to existing cell type identities. To enable cross-comparison, fetal brain, RC, and KH organoid datasets were merged and integrated using CCA (IntegrateLayers()), followed by clustering and dimensionality

reduction on the integrated space. Cell types were assigned using marker gene expression and propagated across datasets. Cluster-specific UMAPs were generated, and composition differences between datasets and conditions were visualized using bar plots, pie charts, and dot plots. Cell-type proportions were calculated using dplyr::group_by() and summarize(), and plotted using ggplot2. Proportions were compared at batch and condition levels. Gene-level comparisons across datasets were shown via FeaturePlot() and DotPlot().

## GO enrichment analysis

To identify differentially expressed genes (DEGs) for each cell type within the dataset, we used the Seurat function FindAllMarkers() with the Wilcoxon test, setting a minimum percentage threshold of 0.2 and a log-fold change threshold of 0.25. This analysis identified marker genes specific to each cluster, which were then saved for further exploration.

For GO enrichment analysis, we performed Biological Process (BP) ontology enrichment using the clusterProfiler package. The marker genes were split by cluster and analyzed with the enrichGO() function, specifying the human genome annotation database (org.Hs.eg.db) as the reference. To account for background gene expression, the entire set of detected genes in the dataset was used as the universe for enrichment testing. The results were corrected for multiple comparisons using the Benjamini–Hochberg (BH) adjustment method, applying a $q$ value cutoff of 0.05. To refine the results and remove redundant GO terms, we applied the simplify() function with a similarity threshold of 0.7, selecting the most representative GO terms based on the lowest adjusted $p$ value. The results were then compiled into a summary data frame and visualized using ggplot2. The dotplot represents the most significantly enriched biological processes, with dot size reflecting the number of genes associated with each GO term and color intensity corresponding to the adjusted $p$ value.

## Metabolic module scoring analysis

To assess metabolic activity in organoid-derived cell populations, we computed module scores for oxidative phosphorylation (OXPHOS) and glycolysis using gene sets associated with their respective GO terms. The analysis was conducted in Seurat, applying the AddModuleScore() function to quantify pathway activation at the single-cell level. For OXPHOS, genes associated with GO:0006119 (oxidative phosphorylation) were retrieved from the org.Hs.eg.db database. Similarly, for glycolysis, genes linked to GO:0006096 (glycolytic process) were extracted. To avoid redundancy, only unique gene symbols were considered for scoring. The module scores were computed for each cell using AddModuleScore() in Seurat, which compares the expression of genes in the pathway against control gene sets. The resulting OXPHOS and glycolysis scores were visualized using UMAP feature plots, highlighting pathway activity across different cell types.

To further analyze metabolic heterogeneity, we generated violin plots comparing module scores across cell types, normalizing y-axis limits to ensure comparability between metabolic states. All computations were performed using the Seurat and ggplot2 packages, ensuring robust visualization and interpretation of metabolic trends in organoid datasets.

## Oxygen consumption rate (OCR) measurements via high-resolution respirometry

To assess mitochondrial function in RC, OCR measurements were performed on 90-day-old organoids using a high-resolution oxygraph (Oroboros O2K). Organoids were maintained in Neurobasal A-Plus medium supplemented with B-27 Plus (2%) and Glutamax (1%) until harvesting. Pools of three organoids were separately dissociated into single-cell suspensions to ensure accurate normalization of OCR values per million cells. Organoids were enzymatically dissociated using a 1:1 mixture of Trypsin (Sigma-Aldrich) and Accutase (StemCell Technologies), supplemented with 10 U/ml DNase I (Thermo Fisher Scientific). The suspension was incubated in the gentleMACS™ Dissociator (Miltenyi Biotec) under the NTDK1 program, followed by filtration through a 70-μm cell strainer to obtain a single-cell suspension. The resulting cells were resuspended in MiR05 buffer, a mitochondrial respiration-optimized medium, to maintain metabolic activity throughout the assay. The metabolic profiling protocol involved a series of sequential additions of specific compounds to dissect the activity of individual mitochondrial complexes and respiratory states. First, 10 μg/ml digitonin (Sigma-Aldrich) was added to permeabilize the plasma membrane, ensuring access to substrates by the mitochondria. Then, 1 mM malate (Sigma-Aldrich), 2.5 mM pyruvate (Sigma-Aldrich), and 10 mM glutamate (Sigma-Aldrich) were added to drive electron flow through complex I (CI) via the tricarboxylic acid (TCA) cycle. To stimulate oxidative phosphorylation (OXPHOS), 2.5 mM ADP (Calbiochem) was introduced, enabling ATP synthesis. Subsequently, 5 mM succinate (Sigma-Aldrich) was added to activate complex II (CII) and assess combined CI + CII OXPHOS activity. To measure the maximal capacity of the electron transport system (ETS), 0.2 mM carbonyl cyanide m-chlorophenyl hydrazone (CCCP) (Sigma-Aldrich), a proton uncoupler, was added stepwise until maximal respiration was achieved. To evaluate the contribution of CI and CII to respiration, 0.5 μM rotenone (Sigma-Aldrich) and 0.5 μM antimycin A (Sigma-Aldrich) were added, respectively. Finally, 2 mM sodium ascorbate (Sigma-Aldrich) and 0.5 mM $N,N,N',N'$-tetramethyl-p-phenylenediamine dihydrochloride (TMPD) (Sigma-Aldrich) were used to stimulate cytochrome c oxidase (complex IV, COX), and 100 mM sodium azide (Sigma-Aldrich) was added to inhibit COX and determine residual oxygen consumption (ROX). These measurements provided a comprehensive metabolic profile of the organoids, enabling the quantification of mitochondrial respiratory capacity, OXPHOS efficiency, and individual contributions of mitochondrial complexes to overall cellular respiration.

## Data availability

The single-cell RNA-sequencing data have been uploaded to Gene Expression Omnibus (GEO) under the reference number GSE306010. The direct URL for the GEO deposition GSE306010: https://www.ncbi.nlm.nih.gov/geo/query/acc.cgi?acc=GSE306010. Confocal images related to Fig. 4 have been deposited in the BioImage archive (S-BIAD2249) and are available at: https://www.ebi.ac.uk/biostudies/bioimages/studies/S-BIAD2249?key=72e757d2-b67f-4ed3-9b95-d6a68c422d0e. The custom Python

codes used for image analysis (necrotic core quantification, organoid morphology, lumen ZO1 quantification, and histological quantification) are available at: https://github.com/ariannaravera/organoids-characterization.

The source data of this paper are collected in the following database record: biostudies:S-SCDT-10_1038-S44319-025-00619-x.

## Peer review information

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

## Acknowledgements

CB was supported by Etat de Vaud, Swiss National Science Foundation N. 310030_215706/1, Italian Telethon Foundation N. GGP20137, Fondazione Italiana per l'Autismo (FIA), PRIN-MUR N. 20227JA8R3. GA was supported by Financement CR-FBM and by CelVivo Aps. BV was supported by a 3RCC PhD fellowship N. DP2022-007. We thank Diane Estade and Manuela Novelli for administrative assistance, Tilmann Achsel and all the members of the Bagni lab for discussions and feedback. We thank Prof. Peng Jin (Emory University) for the hiPSC lines and Dr. Julien Puyal (DNF, University of Lausanne) for advice related to the assessment of the necrotic core. We thank the Lausanne Genomic Technologies Facility (GTF) for single cell RNA sequencing, the Cellular Imaging Facility (CIF) and the Stem Cell and Organoid Facility (SCOF) for support.

## Author contributions

**Giuseppe Aiello**: Conceptualization; Data curation; Software; Formal analysis; Funding acquisition; Investigation; Methodology; Writing—original draft; Writing—review and editing. **Mohamed Nemir**: Data curation; Formal analysis;

Investigation; Methodology; Writing—original draft; Writing—review and editing. **Barbora Vidimova**: Investigation; Writing—review and editing. **Cindy Ramel**: Investigation. **Joanna Viguie**: Investigation. **Arianna Ravera**: Software; Methodology. **Krzysztof Wrzesinski**: Methodology; Writing—review and editing. **Claudia Bagni**: Conceptualization; Resources; Supervision; Funding acquisition; Validation; Methodology; Writing—original draft; Project administration; Writing—review and editing.

Source data underlying figure panels in this paper may have individual authorship assigned. Where available, figure panel/source data authorship is listed in the following database record: biostudies:S-SCDT-10_1038-S44319-025-00619-x.

## Disclosure and competing interests statement

Part of this work was supported by CelVivo ApS. funds to GA. KW is employed at CelVivo ApS. The other authors have no competing or financial interests.

# Expanded View Figures

**Figure EV1. Prolonged aggregation time enhances EBs morphological homogeneity.**

(A) Bar plots showing quantitative comparisons of ten morphological parameters between PA (blue) ($n = 102$, five batches) and RC (yellow) ($n = 88$, three batches) EBs at day 6. Each dot represents an individual EB. The height of the columns represents the median and the error bars represent the interquartile range (i.e., the range between the 25th percentile and the 75th percentile). Statistically significant differences between conditions are indicated. Mann–Whitney test, ns not significant, *$p < 0.05$, ****$p < 0.0001$. Exact $p$ values: area ($p < 0.0001$), perimeter ($p = 0.7908$), average radius ($p < 0.0001$), roundness ($p < 0.0001$), max feret ($p < 0.0001$), min feret ($p < 0.0001$), mean curvature ($p = 0.0353$), std curvature ($p < 0.0001$), std curvature × R0 ($p < 0.0001$), DNE ($p < 0.0001$). (B) Bar plots representing the absolute relative median absolute deviation (MAD) for each morphological parameter, comparing inter-batch variability between PA (blue) ($n = 5$ batches) and RC (yellow) ($n = 3$ batches) conditions at day 6. The height of the columns represents the median and each dot represents one batch, with error bars indicating interquartile ranges. Significant differences are marked. Mann–Whitney test, ns not significant, *$p < 0.05$. Exact $p$ values: area ($p = 0.0357$), perimeter ($p = 0.0357$), average radius ($p = 0.0357$), roundness ($p = 0.0357$), max feret ($p = 0.0357$), min feret ($p = 0.0357$), mean curvature ($p = 0.0357$), std curvature ($p = 0.3929$), std curvature × R0 ($p = 0.0357$), DNE ($p = 0.2500$). (C) Bar plots depicting ten morphological parameters of EBs at day 6, analyzed across two independent iPSC lines (Line 1 and Line 2) cultured under PA (blue) (Line 1:$n = 39$, Line 2:$n = 63$) and RC (yellow) (Line 1:$n = 16$, Line 2:$n = 72$) protocols. Each dot represents an individual EB. The height of the bars represents the mean, and error bars indicate the standard error of the mean (SEM). Statistical significance was assessed using a two-way ANOVA with Šidák's multiple comparisons test, with $p$ values adjusted for multiple comparisons. ns not significant, *$p < 0.05$, **$p < 0.01$, ****$p < 0.0001$. Exact $p$ values: area (PA $p = 0.3743$, RC $p = 0.6160$), perimeter (PA $p = 0.2463$, RC $p = 0.8357$), average radius (PA $p = 0.0340$, RC $p = 0.6476$), roundness (PA $p = 0.0065$, RC $p = 0.8687$), max feret (PA $p = 0.0339$, RC $p = 0.7193$), min feret (PA $p = 0.0941$, RC $p = 0.2902$), mean curvature (PA $p = 0.0071$, RC $p = 0.5519$), std curvature (PA $p = 0.5908$, RC $p < 0.0001$), std curvature × R0 (PA $p = 0.0571$, RC $p = 0.0058$), DNE (PA $p = 0.0042$, RC $p < 0.0001$).

**A**                                    Morphological features

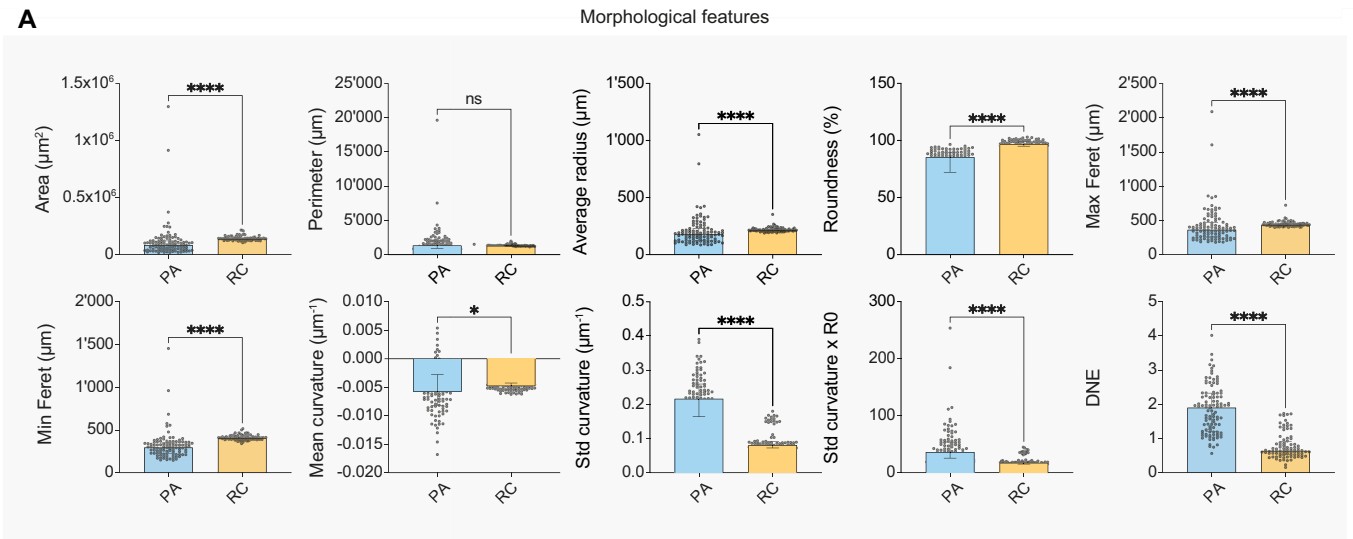

**B**                                    Inter-batch variability

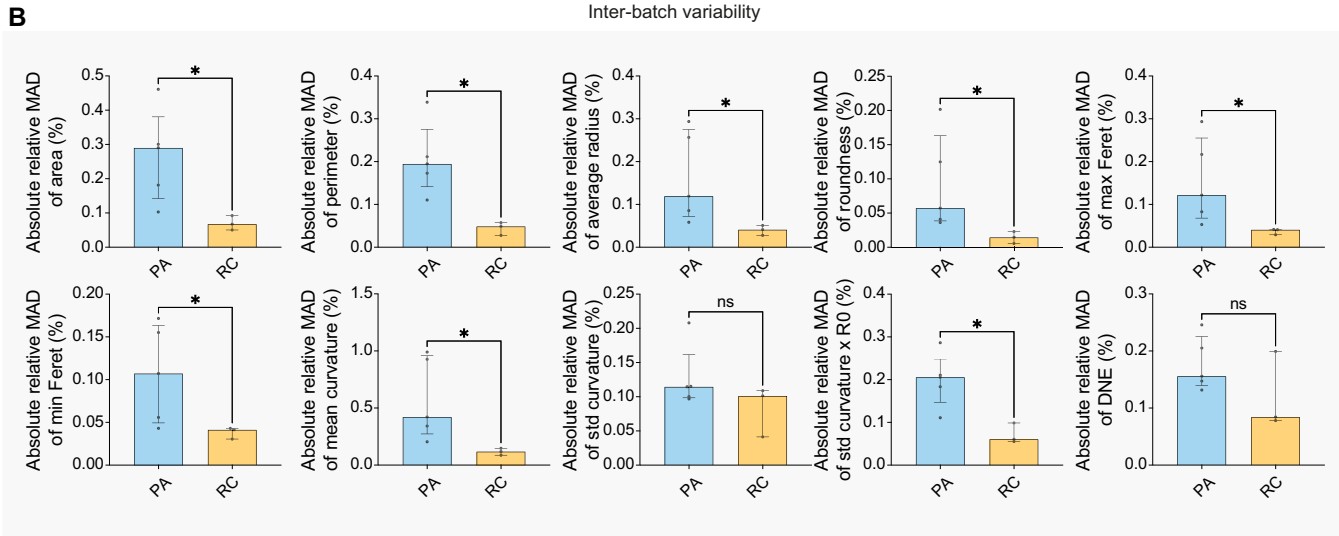

**C**                                    Inter-line variability

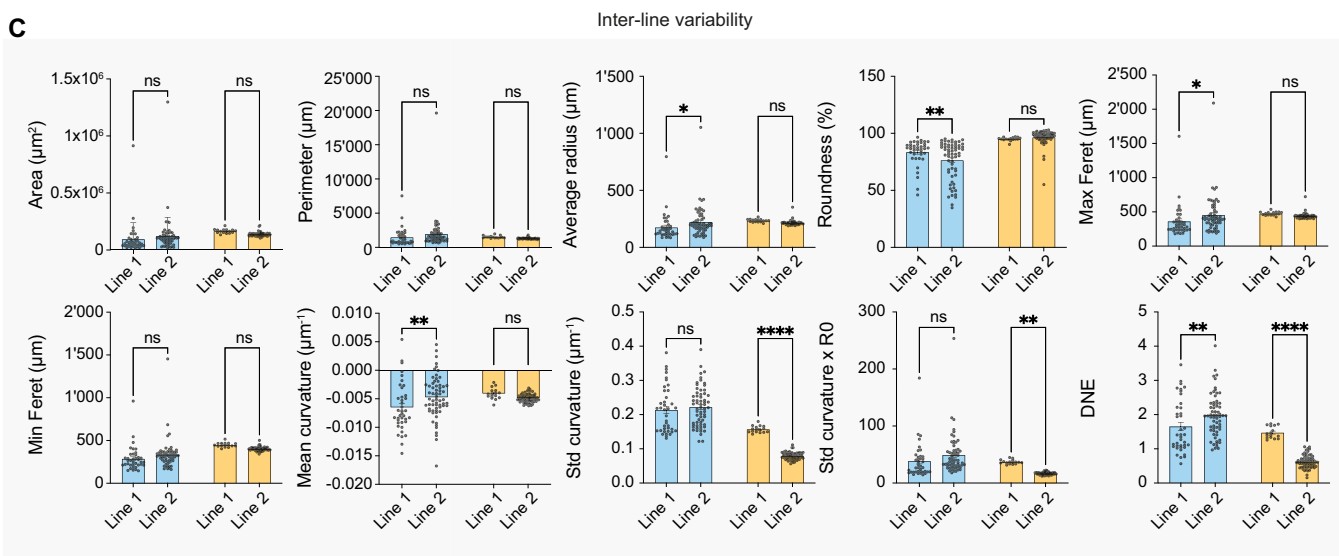

A Morphological features

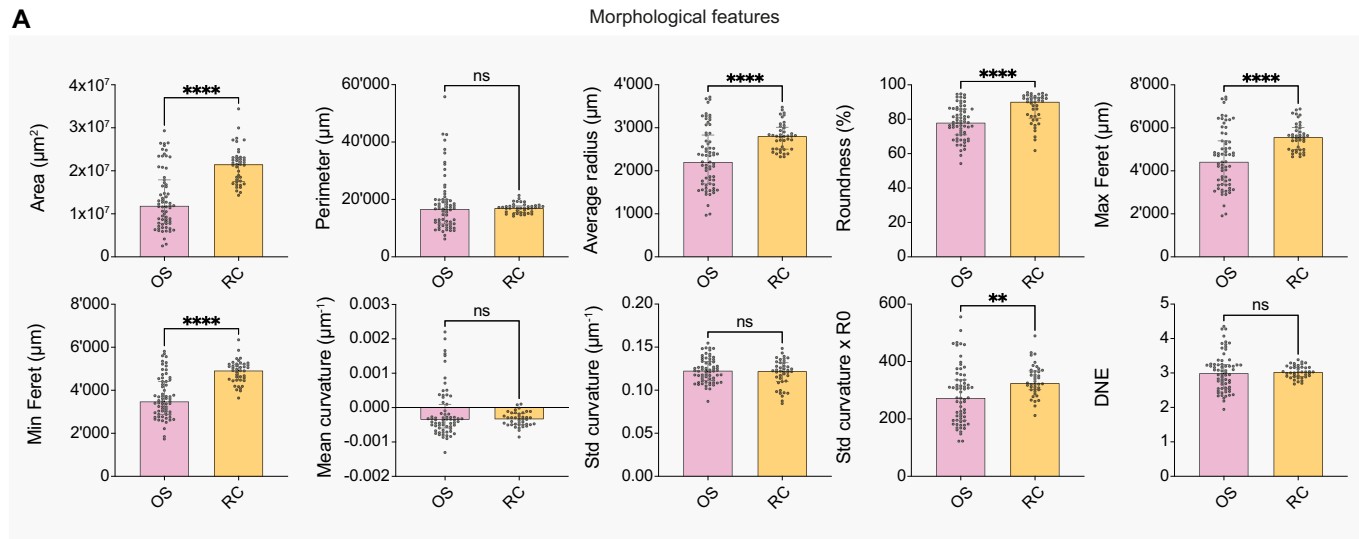

B Inter-batch variability

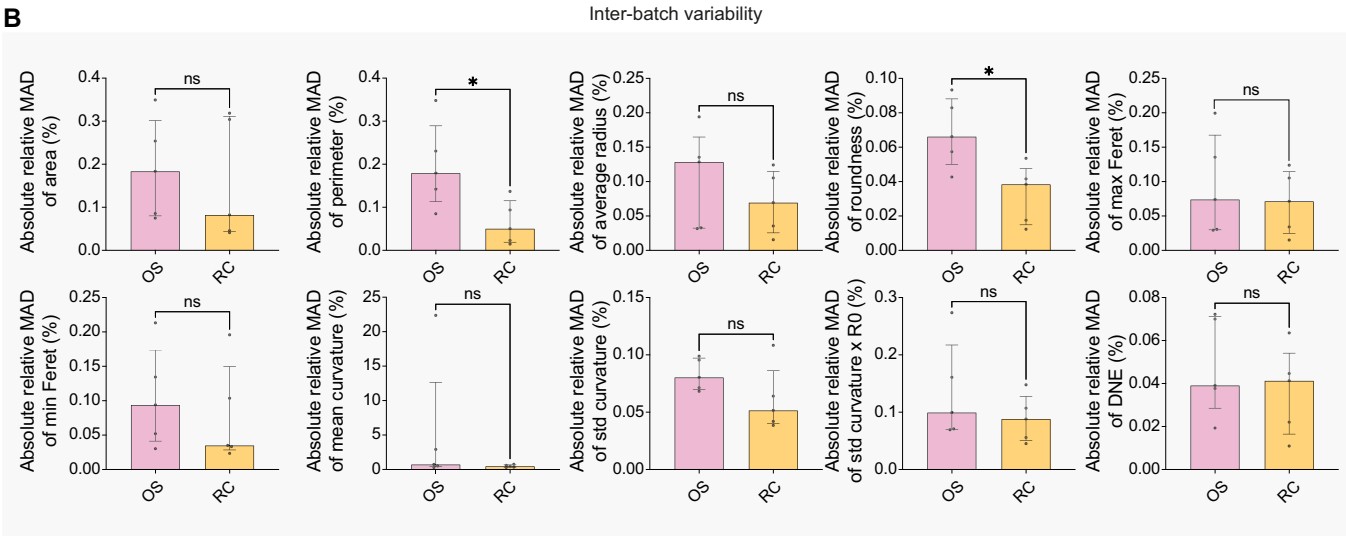

C Inter-line variability

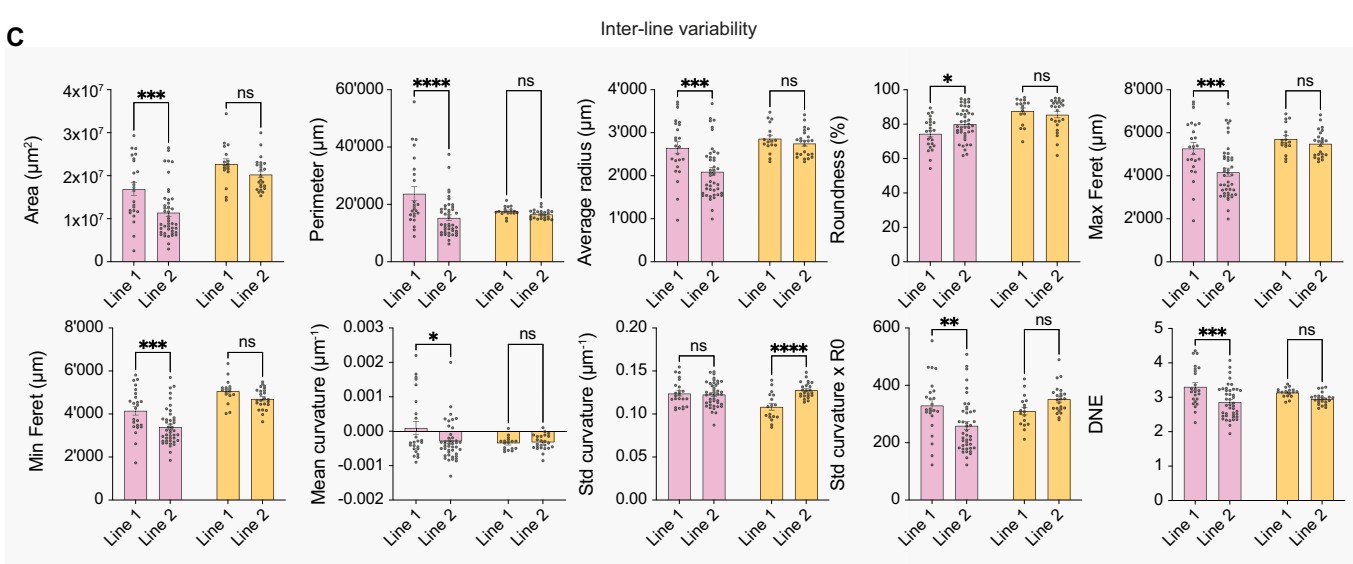

◄ **Figure EV2. RC apparatus improves morphological homogeneity across batches and cell lines.**

(A) Bar plots quantifying ten morphological parameters in OS (pink, $n = 67$, five batches) and RC (yellow, $n = 41$, five batches) organoids at day 90. Each dot represents an individual organoid. The height of the bars represents the median, and the error bars correspond to the interquartile range (i.e., the range between the 25th and 75th percentiles). Statistically significant differences between conditions are indicated. Mann–Whitney test, ns not significant, **$p < 0.01$, ****$p < 0.0001$. Exact $p$ values: area ($p < 0.0001$), perimeter ($p = 0.5366$), average radius ($p < 0.0001$), roundness ($p < 0.0001$), max feret ($p < 0.0001$), min feret ($p < 0.0001$), mean curvature ($p = 0.8205$), std curvature ($p = 0.5038$), std curvature × R0 ($p = 0.0011$), DNE ($p = 0.5366$). (B) Bar plots showing the absolute relative median absolute deviation (MAD) for each morphological parameter, comparing inter-batch variability between OS (pink, $n = 5$ batches) and RC (yellow, $n = 5$ batches) conditions at day 90. The height of the columns represents the median and each dot represents one batch, and error bars indicate interquartile ranges. Significant differences are marked. Mann–Whitney test, ns not significant, *$p < 0.05$. Exact $p$ values: area ($p = 0.5476$), perimeter ($p = 0.0317$), average radius ($p = 0.4206$), roundness ($p = 0.0159$), max feret ($p = 0.6905$), min feret ($p = 0.5476$), mean curvature ($p = 0.3095$), std curvature ($p = 0.1508$), std curvature × R0 ($p = 0.4206$), DNE ($p = 0.6905$). (C) Bar plots depicting ten morphological parameters of organoids at day 90, analyzed across two independent iPSC lines (Line 1 and Line 2) cultured under OS (pink) (Line 1: $n = 24$, Line 2: $n = 43$) and RC (yellow) (Line 1: $n = 17$, Line 2: $n = 24$) protocols. Each dot represents an individual organoid. The height of the bars represents the mean, and error bars indicate the standard error of the mean (SEM). Statistical significance was assessed using a two-way ANOVA with Šidák's multiple comparisons test, with $p$ values adjusted for multiple comparisons. ns not significant, *$p < 0.05$, **$p < 0.01$, ***$p < 0.001$, **** $p < 0.0001$. Exact $p$ values: area (OS $p = 0.0005$, RC $p = 0.3297$), perimeter (OS $p < 0.0001$, RC $p = 0.8638$), average radius (OS $p = 0.0003$, RC $p = 0.7651$), roundness (OS $p = 0.0278$, RC $p = 0.7080$), max feret (OS $p = 0.0002$, RC $p = 0.7798$), min feret (OS $p = 0.0006$, RC $p = 0.2890$), mean curvature (OS $p = 0.0143$, RC $p = 0.9863$), std curvature (OS $p = 0.9250$, RC $p < 0.0001$), std curvature × R0 (OS $p = 0.0018$, RC $p = 0.2055$), DNE (OS $p = 0.0001$, RC $p = 0.3061$).

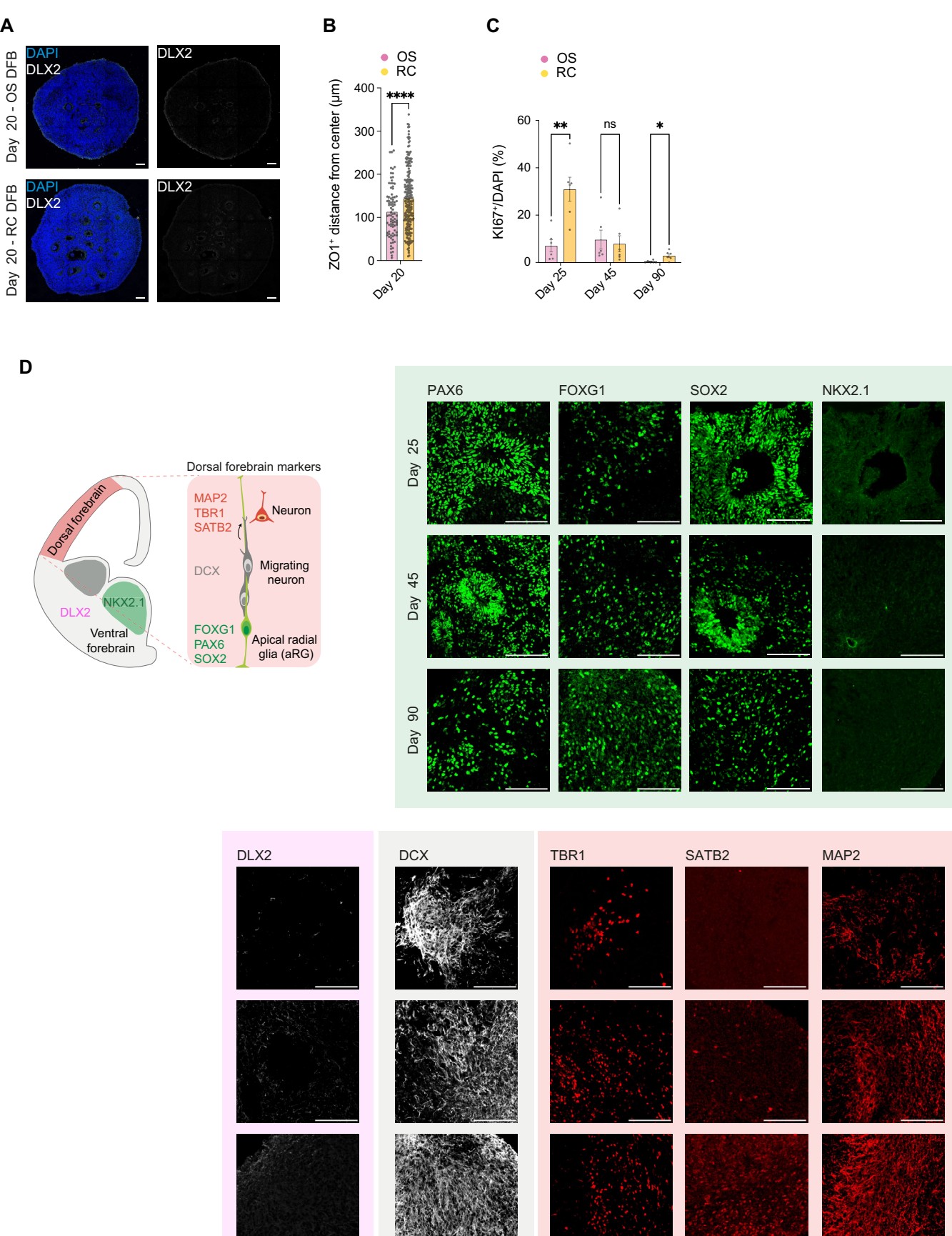

◀ **Figure EV3. RC organoids do not show expression of ventral forebrain markers.**

(A) Immunofluorescence images for DLX2 (white) and DAPI (blue) in OS and RC organoids at day 20. Scale bars: 100 µm. (B) Quantification of of ZO1⁺ rosette distance from the organoid center. Each dot represents a single ZO1⁺ rosette ($n = 3$ batches per line). Data were presented as mean ± SEM. Statistical analysis was performed using unpaired two-tailed Welch's *t*-test with correction for multiple comparisons using the two-stage step-up method of Benjamini, Krieger, and Yekutieli. ****$p < 0.0001$. Exact *p* value: $p = 0.000104$. (C) Quantification of the proportion of KI67⁺/DAPI⁺ cells across time points ($n = 3$ batches per line). Data were presented as mean ± SEM. Statistical analysis was performed using an unpaired two-tailed Welch's *t*-test with correction for multiple comparisons using the two-stage step-up method of Benjamini, Krieger, and Yekutieli. ns not significant, *$p < 0.05$, **$p < 0.01$. Exact *p* values: day 25 ($p = 0.003616$), day 45 ($p = 0.746061$), day 90 ($p = 0.020226$). (D) Schematic representation of dorsal and ventral forebrain markers, illustrating marker expression patterns in organoids. Immunofluorescence staining at days 25, 45, and 90, showing expression of PAX6, FOXG1, SOX2, and NKX2.1 (green), DCX (white), TBR1, SATB2, and MAP2 (red) and DLX2 (violet) across different time points. Scale bars: 100 µm.

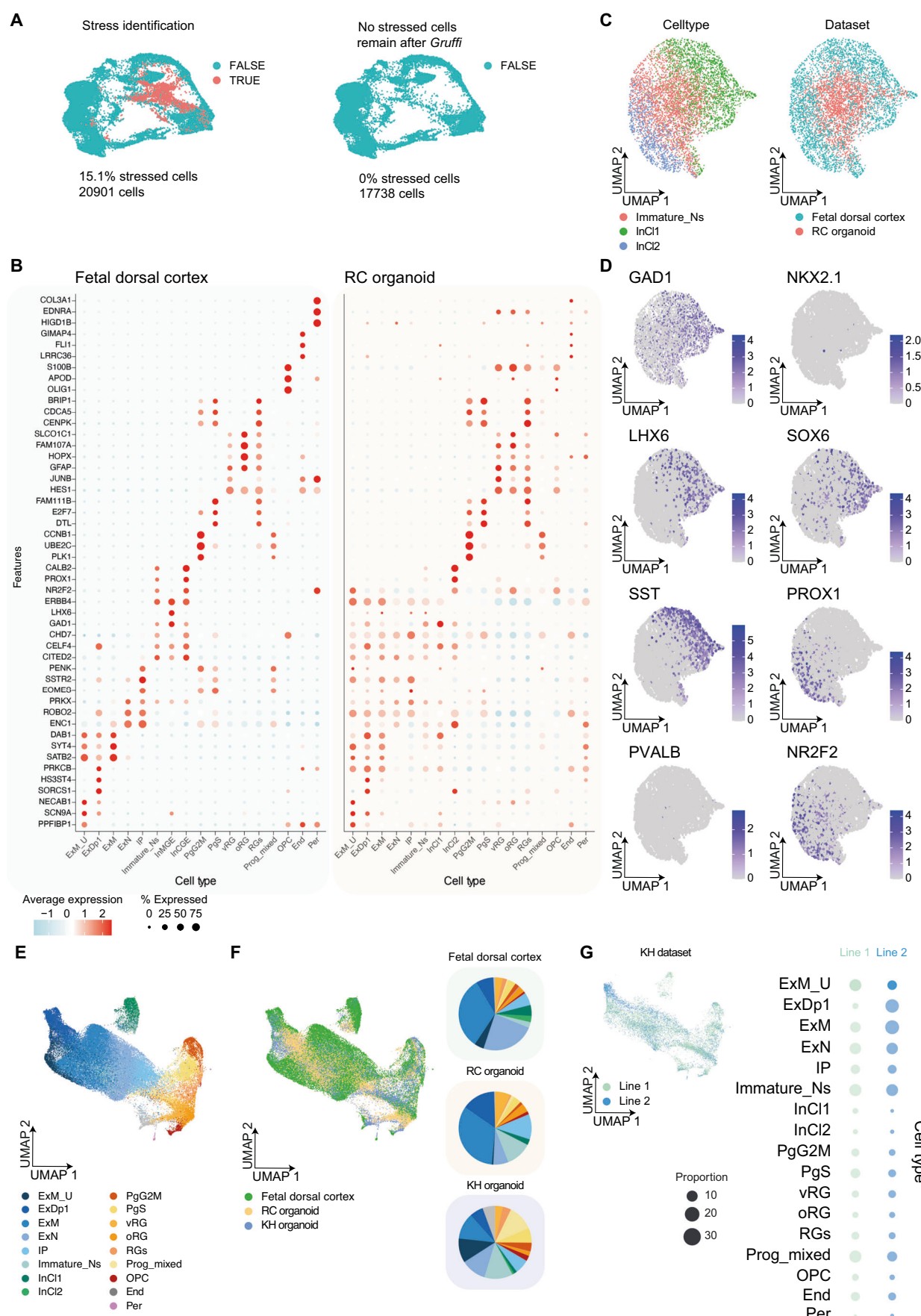

◀ **Figure EV4. Quality control and integrative transcriptomic analysis of RC organoids.**

(A) Identification and removal of stressed cells using the GRUFFI algorithm (Vertesy et al, 2022). The left UMAP plot displays the proportion of stressed cells (red) in the dataset before filtering, with 15.1% of cells (20,901 total) identified as stressed. The right UMAP plot shows the dataset after GRUFFI filtering, where all stressed cells have been removed, retaining 17,738 high-quality cells for downstream analysis. (B) Dotplot showing the expression of canonical marker genes across annotated cell types in the fetal cortex (left) and RC organoids (right), confirming similar transcriptional patterns between corresponding populations. Dot size represents the percentage of expressing cells; color intensity indicates average expression. (C) UMAP representations highlighting the three interneuron-related clusters: Immature_Ns, InMGE, and InCGE. Cells are colored by cell type (left) or dataset of origin (right), showing separation between fetal and organoid-derived interneurons. (D) Feature plots of selected interneuron markers (GAD1, NKX2.1, LHX6, SOX6, SST, PROX1, PVALB, and NR2F2). (E) UMAP of the integrated dataset including fetal cortex (Polioudakis et al, 2019), RC organoids, and KH organoids (Khan et al, 2020). (F) Right: UMAP of the integrated dataset color-coded by dataset of origin. Left: Pie charts indicating the proportional distribution of different cell types in each dataset. (G) Proportional representation of cell types across the two KH organoid lines, visualized through dot plots. Cell types include: ExM_U (maturing excitatory upper enriched), ExDp1 (excitatory deep layer 1), ExM (maturing excitatory), ExN (migrating excitatory), IP (intermediate progenitors), Immature_Ns (immature neurons), InCl1 (interneuron cluster 1), InCl2 (interneuron cluster 2), PgG2M (cycling progenitors in G2/M phase), PgS (cycling progenitors in S phase), vRG (ventricular radial glia), oRG (outer radial glia), RGs (radial glial populations), Prog_mixed (mixed progenitors), OPC (oligodendrocyte progenitor cells), End (Endothelial cells) and Per (pericytes).

**A**

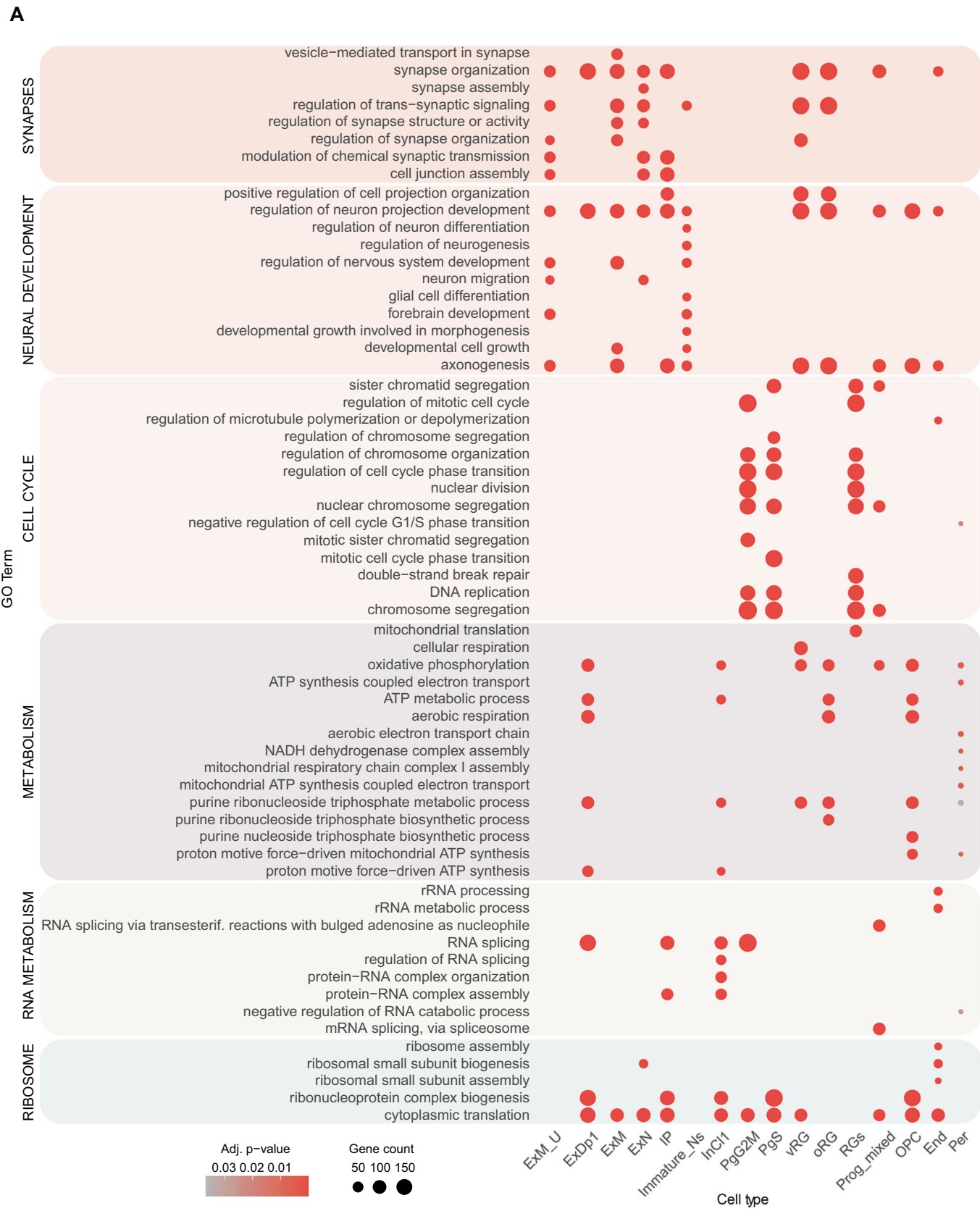

◀ **Figure EV5. Functional annotation.**

(A) Gene Ontology (GO) enrichment analysis of biological processes across different cell types in the dataset. The dot size corresponds to the number of genes enriched within each GO term, while the color intensity reflects the adjusted *p* value, indicating the significance of enrichment. Cell types include: ExM_U (maturing excitatory upper enriched), ExDp1 (excitatory deep layer 1), ExM (maturing excitatory), ExN (migrating excitatory), IP (intermediate progenitors), Immature_Ns (immature neurons), InCl1 (interneuron cluster 1), InCl2 (interneuron cluster 2), PgG2M (cycling progenitors in G2/M phase), PgS (cycling progenitors in S phase), vRG (ventricular radial glia), oRG (outer radial glia), RGs (radial glial populations), Prog_mixed (mixed progenitors), OPC (oligodendrocyte progenitor cells), End (Endothelial cells) and Per (pericytes). Differentially expressed genes for each cluster were identified using the Wilcoxon rank-sum test (Seurat FindAllMarkers function). GO enrichment analysis was performed with the clusterProfiler package (enrichGO function) using the Benjamini–Hochberg method for multiple testing correction (*q* value cutoff = 0.05).

