## [Peer Review File · EMBO Reports]

Increased Reproducibility of Brain Organoids through Controlled Fluid Dynamics

Giuseppe Aiello, Mohamed Nemir, Barbora Vidimova, Cindy Ramel, Joanna Viguié, Arianna Ravera, Krzysztof Wrzesinski, and Claudia Bagni

Corresponding author(s): Claudia Bagni (claudia.bagni@unil.ch)

Review Timeline:

Submission Date:	31st Mar 25
Editorial Decision:	2nd May 25
Revision Received:	13th Aug 25
Editorial Decision:	30th Sep 25
Revision Received:	10th Oct 25
Accepted:	16th Oct 25

Editor: Esther Schnapp

Transaction Report:

Dear Claudia,

Thank you for the submission of your manuscript to EMBO reports. We have now received the full set of referee reports that is pasted below.

As you will see, the referees acknowledge that the findings are potentially interesting. However, they also have several comments and suggestions for how the study could be improved. I think all comments are good and should in principle be addressed, but please let me know in case you disagree, and we can discuss the exact revision requirements further, also in a video chat, if you like.

I would thus like to invite you to revise your manuscript with the understanding that the referee concerns must be fully addressed and their suggestions taken on board. Please address all referee concerns in a complete point-by-point response. Acceptance of the manuscript will depend on a positive outcome of a second round of review. It is EMBO reports policy to allow a single round of major revision only and acceptance or rejection of the manuscript will therefore depend on the completeness of your responses included in the next, final version of the manuscript.

We realize that it is difficult to revise to a specific deadline. In the interest of protecting the conceptual advance provided by the work, we recommend a revision within 3 months (2nd Aug 2025). Please discuss the revision progress ahead of this time with the editor if you require more time to complete the revisions.

- 1) A data availability section providing access to data deposited in public databases is missing. If you have not deposited any data, please add a sentence to the data availability section that explains that.
- 2) Your manuscript contains statistics and error bars based on $n=2$. Please use scatter blots in these cases. No statistics should be calculated if $n=2$.

5) a complete author checklist, which you can download from our author guidelines <https://www.embopress.org/page/journal/14693178/authorguide>. Please insert information in the checklist that is also

reflected in the manuscript. The completed author checklist will also be part of the RPF.

6) Please note that all corresponding authors are required to supply an ORCID ID for their name upon submission of a revised manuscript (<<https://orcid.org/>>). Please find instructions on how to link your ORCID ID to your account in our manuscript tracking system in our Author guidelines <<https://www.embopress.org/page/journal/14693178/authorguide#authorshipguidelines>>

7) Before submitting your revision, primary datasets produced in this study need to be deposited in an appropriate public database (see <https://www.embopress.org/page/journal/14693178/authorguide#datadeposition>). Please remember to provide a reviewer password if the datasets are not yet public. The accession numbers and database should be listed in a formal "Data Availability" section placed after Materials & Method (see also <https://www.embopress.org/page/journal/14693178/authorguide#datadeposition>). Please note that the Data Availability Section is restricted to new primary data that are part of this study. * Note - All links should resolve to a page where the data can be accessed. *
If your study has not produced novel datasets, please mention this fact in the Data Availability Section.

12) All Materials and Methods need to be described in the main text using our 'Structured Methods' format, which is required for all research articles. According to this format, the Methods section includes a Reagents and Tools Table (listing key reagents, experimental models, software and relevant equipment and including their sources and relevant identifiers) followed by a Methods and Protocols section describing the methods using a step-by-step protocol format. The aim is to facilitate adoption of the methodologies across labs. More information on how to adhere to this format as well as a downloadable template (.docx) for the Reagents and Tools Table can be found in our author guidelines: <https://www.embopress.org/page/journal/14693178/authorguide#structuredmethods>.

An example of a Method paper with Structured Methods can be found here: <https://www.embopress.org/doi/full/10.1038/s44320-024-00037-6#sec-4>

We would also welcome the submission of cover suggestions, or motifs to be used by our Graphics Illustrator in designing a

cover.

I look forward to seeing a revised form of your manuscript when it is ready. Please use this link to submit your revision:

Best wishes,
Esther

Referee #1:

The manuscript by Aiello and colleagues aims, by controlling fluid dynamics, to improve brain organoid reproducibility formed from ECM-free protocols. Reproducibility indeed remains a persistent issue in the field. The authors characterized their method by extensively analyzing morphological features of the organoids, across two PSC lines, and characterized the organoids by some markers/single-cell RNA seq. While the morphological improvements of the brain organoids are clearly more consistent, it remains uncertain how this translates to their phenotypes. The phenotypic characterization is in its current stage too shallow, lacks characterization for batch-to-batch/within batch variability, and comparisons with the existing protocols are needed to make these points stronger. This would be needed for researchers aiming to implement this protocol in their work.

My major comments and questions are as follows:

1. Introduction: The authors emphasize that use of Matrigel affects morphology and introduces variability in differentiation outcomes. While it is clear that the inclusion of exogenous ECM can influence morphology and differentiation of brain organoids, I do not recall that the cited works consistently show that Matrigel specifically introduces variability, particularly if the authors are referring to variability between different batches. For example, it appears to aid consistency to form neuroepithelial buds, etc. Refer to Pagliaro et al. PMID: 39826996 discussing these studies. Could the authors further clarify what is meant, is it the impact of exogenous ECM alone, or is it a combination of factors? Additionally, please clarify whether this concerns batch-to-batch variability or variability within the same experiment.
2. In the first section of the results, the authors mention "velocity and shear stress analysis confirmed stable flow dynamics across constructs with different densities". Could the authors mention either here or in the Method section what is meant by "different densities"
3. The authors compare in the main body of the text two protocols for generating brain organoids, referred to as 'PA' (adapted from Sloan et al., 2018) and 'RC' (their novel method). While I fully agree that the differences observed at Day 6 are quite evident, I believe these differences are primarily due to the use of AggreWell plates, as opposed to the spontaneous EB formation seen in the 'PA' protocol. Therefore, the 'OS' protocol appears more appropriate for comparisons, and indeed less striking differences are observed, with overall quite similar heterogeneity, this means that improvements must come from phenotypic superiority, which is missing in the current form of this manuscript.
4. Please merge some of the data (including schematics) from EV1/2 into main Figure 1, comparing simultaneously the RC, OS, and PA protocols. Please also include clear information about the organoid generation method for the 'OS' protocol in the Methods section.
5. Related to this point, the schematic overview of the different protocols in Figures EV1 and EV2 helps to illustrate how they differ in the differentiation process of the organoids. However, it is unclear from these schematics what the differences are in the aggregation step. Please edit the figures and the Methods section accordingly to make this more intuitive.
6. In Figure 3B, the PAX6+ cells appear located within only the inner rosettes of the organoids. What is the identity of the rest of the cells (that is also DCX-). Additionally, more markers should be used to confirm cortical identity (e.g. EMX2). Is similar patterning observed both the 'PA' and 'OS' protocols? Please provide comparative stainings (whole organoid and zoom-ins) and

including quantifications of rosette positivity across organoids from both lines used.

7. Related to the previous point, the positivity (and layering, and co-staining for e.g. stem/differentiated progeny) for the different markers shown in Figure 3 should be significantly extended. This should be shown across different organoids and different batches, with associated quantifications, in order to strengthen the observation of improved reproducibility using the "RC" protocol. Comparative stainings across the 'PA' and 'OS' protocols (with quantifications) are needed as well to provide phenotypic comparisons.

8. RC organoids appear overall bigger (area, EV2), are the stem cells more proliferative?

9. The single-cell RNA sequencing data is presented in a too superficial way. While cell classes are annotated, few markers are shown to substantiate this annotation. An extensive dot plot with markers for the different cell types should be shown to support their presence (exM, exN, exDP1, etc.). Consider also integrating with existing previous datasets of the comparative protocols used.

10. After day 25, the organoids generated using the "RC" protocol are moved back on to an orbital shaker. Why is this choice made? Does the fFSS not influence the organoids at this point? This could be of interest to show.

11. Related to the previous point, the differences between the fFSS of the orbital shaker versus the rotating chamber should be shown.

Referee #2:

The manuscript presents an important technical advancement in brain organoid culture, offering a compelling case that a low-shear rotating chamber improves reproducibility in organoid development. The work is timely, technically well-executed, and broadly relevant. The conclusions are generally well-supported, although several aspects of the experimental design and data presentation require clarification or reframing. The report below highlights specific points that need to be addressed before the manuscript can be considered for publication.

Major Comments

1. Comparison with Orbital Shaker in CFD Analysis (Figure 1)

The manuscript claims that the RC minimizes fFSS, but Figure 1 only includes CFD simulations for the RC. Since the RC is compared to an orbital shaker in later sections, it would be beneficial to show CFD or at least cite data on shear stress levels in an orbital shaker. This would better justify the claim that RC's low-shear environment is unique.

2. RC vs PA Comparison at Day 6 is Misleading

The RC protocol is compared to the "PA" protocol at day 6, but the RC device is not yet in use at this stage. The observed morphological differences at this point are due to the aggregation method (AggreWell vs static), not the rotating chamber. This comparison should be reframed or explicitly stated to be a comparison of aggregation methods.

3. No Data Beyond Day 6 for PA Protocol

The schematic implies PA organoids are cultured to day 90, but only day 6 data are shown. This is misleading. Either provide later data for the PA protocol or revise the schematic and text to reflect that the PA comparison is limited to early aggregation.

4. Limited Justification for Using RC Only During Days 6-25

The RC is used only during neural induction. No rationale is given for discontinuing RC culture at day 25. Considering its apparent benefits, why not continue its use during maturation? If there are technical limitations, these should be acknowledged.

5. No Marker or Cell Composition Data for Orbital Shaker Protocol

Although the manuscript compares morphological outcomes between RC and orbital shaker protocols, it does not include scRNA-seq or immunostaining for the orbital shaker condition. Since the claim is that RC improves reproducibility, it would be valuable to show whether this applies to cellular composition and tissue identity as well.

6. Gruffi Filtering Requires Justification

The authors use Gruffi to remove stress-affected cells from the scRNA-seq dataset. This is a non-standard approach. The

rationale should be explained, and the impact of this filtering on inter-line comparisons should be discussed.

7. Justification of Reference Dataset (Polioudakis et al.)

The authors use a mid-gestation fetal brain dataset (GW17-18) for annotation. They should explain why this is an appropriate reference for day 90 organoids.

8. Metabolic Profiling is Interesting but Shallow

The OCR analysis reveals increased OXPHOS in RC organoids, but the metabolic phenotype is not deeply characterized. No data on ATP, ROS, or glycolysis are presented. These could contextualize whether elevated OXPHOS reflects better health, more mature neurons, or simply more mitochondria. At a minimum, the authors should discuss these limitations.

9. Discrepancy Regarding the Presence of InMGE Cells

In Figure 4B, the scRNA-seq analysis shows a visible population of InMGE cells (interneuron progenitors from the medial ganglionic eminence) in RC organoids. However, the text states that ventral forebrain markers were undetectable, reinforcing dorsal identity. This discrepancy should be clarified. If InMGE cells were identified based on transcriptional similarity, the authors should acknowledge this and discuss whether these may represent real ventral differentiation events, misclassification, or noise. Clarification is essential to support the claim of strict dorsal patterning.

Minor Comments

1. Terminology

Define technical terms like "Dirichlet Normal Energy" in biological terms to aid comprehension.

2. Protocol Definitions

Clarify differences between PA, OS, and RC protocols. Ensure the timeline and use of dynamic/static conditions are clearly and consistently described.

3. Data Integration

The authors should mention the possibility of comparing their data with other published scRNA-seq datasets (e.g., Velasco et al., Øhlenschläger et al., Saglam-Metiner et al.) to contextualize findings.

4. Schematic Clarifications

In figures or text, indicate explicitly that PA organoids were not cultured beyond day 6.

Referee #3:

In this manuscript, Aiello et al. make a case for improving brain organoid generation by minimizing fluid flow shear stress, which can impact organoid development and consistency. Using a new vertical rotating chamber (RC) system, they show better reproducibility across batches and iPSC lines, a metabolic shift toward oxidative phosphorylation, and more controlled neural fate acquisition that closely models human cortical development. The work adds valuable insight to the field, but a few important points need to be addressed for the findings to be fully convincing:

1/ A major goal of this article is to generate organoids with a reproducible shape. While this is indeed a useful goal, a caveat here is that high quality organoid batches are often not round, but with prominent lobes corresponding to ventricular-like structures that stick out of the organoid. This is precisely the point of Chiaradia et al, 2023, where it is shown that organoids with more complex morphologies better mimic in vivo fetal brain development. While this article is cited, this apparent conflict should be clearly stated. Nevertheless, protocols for organoids with more regular and therefore reproducible shapes can indeed be needed for certain applications and this does fill a need in the field. All these considerations should be discussed.

2/ Related to this, on page 7, the authors state that the organoids maintain a consistent spherical shape during differentiation. However, by day 45, the organoids appear to exhibit a more irregular morphology, which contradicts this claim. The authors should address these differences and clarify whether the observed morphological changes are representative or due to experimental variation.

3/ The presentation and text for Figure 2 should be simplified and clarified. In figures 2D, E, the authors compare EB generation methods and analyze results early, at Day 6, while in figure 2F, 2G they compare agitation methods, in both cases with their EB

formation assay, and analyze results at day 90. The text should be clarified, simplified and shortened to clearly describe what the 2 variables are in these 2 experiments. Also, the OS protocol could be better described.

4/ Aiello et al. state that the RC protocol "preserves both the external morphology and the inner cytoarchitecture" of organoids. However, the neuroepithelial rosettes, which are highlighted as a key feature of early neural patterning, are reported to localize predominantly in the innermost regions of the organoids. Could the authors elaborate on why these structures are not also observed at the periphery? Is this distribution expected for dorsal forebrain patterning at day 20, or could it indicate limitations in nutrient diffusion, polarity establishment, or differentiation gradients within the organoid?

5/ Hypoxia is often observed at the center of the organoids, and imaging-based studies often therefore concentrate on the outside regions. Given that here the lumen are concentrated in the center, the authors should verify the levels of hypoxia and cell death in these regions.

6/ In Figure 3B, the DAPI staining shows the presence of some PAX6-negative neuroepithelial rosettes. Could the authors elaborate on the identity of these structures? Are they non-neural, ventral, or PAX6-negative neural populations? Further clarification would help support the claim of consistent dorsal forebrain patterning.

7/ Furthermore, to better support the claim of preserved cytoarchitecture, the authors are encouraged to expand the panel of markers and quantify rosette formation (e.g., using ZO-1 staining for apical junctions). Figure 3D gives the impression that rosettes are disappearing earlier than with classical protocols.

8/ In Figure 3B, DCX-positive cells are predominantly located at the periphery of the organoids, while PAX6-positive neuroepithelial rosettes are localized to the innermost regions. If these rosettes represent neural progenitor zones, it is unclear why newly differentiated immature neurons (DCX+) appear at such a distance. Could the authors clarify whether this spatial separation is consistent with expected neurogenic progression and radial migration, or if it might suggest a disruption in organization? To better support the rosette architecture and demonstrate appropriate fate acquisition, the authors are encouraged to include additional pan-neuronal markers such as NeuroD2, NeuN or HuC/D alongside progenitor markers (e.g., Pax6 or Sox2). Co-staining would allow for a more detailed spatial and temporal analysis of progenitor-to-neuron transition within the organoid.

Minor points

1/ Could the authors elaborate on the CFD analysis in the results section (page 6)? Similarly, could the authors provide a scale range for shear forces, to clarify what is here considered as low and high forces. What is the actual range of shear stress values observed in the RC versus compared systems? Also, the criteria used to define these thresholds is missing and could help in improving interpretation.

2/ The authors claimed that "The shear stress Levels remained low across all tested conditions"; however, the manuscript does not clearly define what these conditions were. Could the authors clarify this point?

3/ On page 8, the authors describe RC EBs as both more uniform and exhibiting slightly higher fluctuations in standard curvature and DNE (Fig EV1E). This appears somehow contradictory, and some clarification is needed.

4/ The authors report the presence of interneuron populations from both the medial (InMGE) and caudal (InCGE) ganglionic eminences based on scRNA-seq analysis. However, they also state that the ventral forebrain marker NKX2.1 was not expressed, reinforcing a dorsal forebrain identity. This presents a potential contradiction, as NKX2.1 is a critical and well-established marker of MGE-derived interneurons. The authors should clarify how InMGE populations were identified in the absence of NKX2.1 expression. The authors are encouraged to cite articles identifying dorsally born interneurons.

5/ In the last figure Aiello et al. explore mitochondrial metabolism in RC organoids, this is an important addition to the field. The authors show that scRNA-seq data suggests increased OXPHOS gene expression in progenitor and radial glial populations, while glycolysis genes are more uniformly but weakly expressed. However, mRNA levels do not always correlate with metabolic activity. The authors should acknowledge this limitation and clarify how well these transcriptional signatures are expected to translate to functional energy states.

6/ Finally, the authors discuss the benefits of the RC system over other existing approaches, particularly mitigating fFSS during early developmental stages. However, it remains unclear why RC protocol is currently limited to early time points, while organoids are later transferred to conditions that reintroduce mechanical forces. Given that brain organoids continue to develop and mature over extended culture periods, the authors are encouraged to discuss the potential for extending the RC system to later stages. Could this system be adapted or scaled to support prolonged culture while still maintaining minimal fFSS?

1 Point by Point**2 Increased Reproducibility of Brain Organoids through Controlled Fluid Dynamics**

Giuseppe Aiello, Mohamed Nemir, Barbora Vidimova, Cindy Ramel, Joanna Viguie, Arianna Ravera,
Krzysztof Wrzesinski and Claudia Bagni

We sincerely thank the reviewers and the editor for their thorough evaluation of our manuscript and
for their constructive and insightful comments that helped to significantly improve the clarity and
impact of our work. We have carefully considered each point raised and extensively revised the
manuscript accordingly. All modifications in the manuscript are highlighted in **blue**, and we refer to
specific **line numbers** in the revised version.

Referee #1:

The manuscript by Aiello and colleagues aims, by controlling fluid dynamics, to improve brain
organoid reproducibility formed from ECM-free protocols. Reproducibility indeed remains a
persistent issue in the field. The authors characterized their method by extensively analyzing
morphological features of the organoids, across two PSC lines, and characterized the organoids by
some markers/single-cell RNA seq. While the morphological improvements of the brain organoids are
clearly more consistent, it remains uncertain how this translates to their phenotypes. The phenotypic
characterization is in its current stage too shallow, lacks characterization for batch-to-batch/within
batch variability, and comparisons with the existing protocols are needed to make these points
stronger. This would be needed for researchers aiming to implement this protocol in their work.

My major comments and questions are as follows:

1. Introduction: The authors emphasize that use of Matrigel affects morphology and introduces
variability in differentiation outcomes. While it is clear that the inclusion of exogenous ECM can
influence morphology and differentiation of brain organoids, I do not recall that the cited works
consistently show that Matrigel specifically introduces variability, particularly if the authors are
referring to variability between different batches. For example, it appears to aid consistency to form
neuroepithelial buds, etc. Refer to Pagliaro et al. PMID: 39826996 discussing these studies. Could the
authors further clarify what is meant, is it the impact of exogenous ECM alone, or is it a combination
of factors? Additionally, please clarify whether this concerns batch-to-batch variability or variability
within the same experiment.

We thank the reviewer for highlighting this point and for suggesting to refer to Pagliaro *et al*, 2025
(PMID: 39826996), which indeed clarifies the nuanced role of exogenous ECM components like
Matrigel in brain organoid culture. We have now revised the Introduction and the Discussion sections
to better clarify our statements concerning the use of ECM and also cited Pagliaro et al.

Specifically, we now state in the introduction that the heterogeneous composition of Matrigel can
potentially affect differentiation fidelity, given that Matrigel's lots vary in their biochemical properties
(Aisenbrey & Murphy, 2020 PMID 32953138). Despite facilitating initial morphological organization
and neuroepithelial formation, this "undefined" ECM may affect endogenous ECM production,
potentially influencing developmental fidelity and contributing to variability in differentiation
outcomes across experiments as reported in (Jain *et al.*, 2025 PMID: 40533563; Long *et al.*, 2018
PMID: 30078576 ; Martins-Costa *et al.*, 2023 PMID: 37842725, Pagliaro *et al.*, 2025 PMID:
39826996). We have now added this part in the Introduction (lines 70-78). We thought it was also
important to cite that novel synthetic or naturally derived alternatives are under development, aiming
at obtaining more standardized and compositionally relevant ECM scaffolds. These new reagents
could significantly benefit the reproducibility and fidelity of human brain organoid models. We have
included this comment in the Discussion (lines 465-472).

2. In the first section of the results, the authors mention "velocity and shear stress analysis confirmed
stable flow dynamics across constructs with different densities". Could the authors mention either
here or in the Method section what is meant by "different densities".

We thank the referee for this comment and lack of clarity. CFD simulations were performed using a
Navier-Stokes-based solver to model fluid velocity and shear stress for constructs with different
density related to compactness of the construct affecting relative weight/buoyancy of constructs with

the same surface area (relative mass 1010, 1020, and 1030 where 1000 is density/relative mass of
surrounding medium, generated in COMSOL Multiphysics 6.2 by Resolvent Denmark PS, Maaloev,
Denmark). Shear force distribution was computed based on the chamber's geometry while the
velocity profiles were extracted to evaluate the stability of the system. Data were plotted to compare
velocity and shear stress from drag force across different conditions. We clarified this part in the
Methods (lines 532-540).

3. The authors compare in the main body of the text two protocols for generating brain organoids,
referred to as 'PA' (adapted from Sloan et al., 2018) and 'RC' (their novel method). While I fully agree
that the differences observed at Day 6 are quite evident, I believe these differences are primarily due
to the use of AggreWell plates, as opposed to the spontaneous EB formation seen in the 'PA' protocol.
Therefore, the 'OS' protocol appears more appropriate for comparisons, and indeed less striking
differences are observed, with overall quite similar heterogeneity, this means that improvements must
come from phenotypic superiority, which is missing in the current form of this manuscript.

We thank the reviewer for highlighting this important point. As correctly pointed out by the reviewer,
given the importance of early morphogenetic control on reproducibility (Gjorevski *et al.*, 2022 PMID:
34990240) we also questioned the effect of keeping the organoids on AggreWell for short (1 day) or
longer times (6 days) and therefore checked the possible effects on the morphology of the organoids.

To better clarify that PA EBs were cultured only to test the impact of early aggregation, we updated
the Methods section for the PA protocol (Methods, lines 522-525) and changed the schematic in **Fig**
**2D** (top panel).

Our results suggested that longer aggregation produces more reproducible early EBs (**Fig. 2D**
bottom).

Building on this finding and in agreement with the reviewer's suggestion, we compared the organoids
obtained using the RC and the OS protocol, morphologically (**Fig 2G-I, EV2**), histologically (**new**
**Fig 3, EV3**) and metabolically (**Fig 5**). The protocols are now clarified and better explained.

4. Please merge some of the data (including schematics) from EV1/2 into main Figure 1, comparing
simultaneously the RC, OS, and PA protocols. Please also include clear information about the
organoid generation method for the 'OS' protocol in the Methods section.

We apologize for the confusion. We applied the PA protocol to assess the initial aggregation length
under static suspension conditions, and compared it to the RC protocol, which utilized AggreWells for
6 days. For the OS protocol, the aggregation phase was carried out as in the RC protocol, whereas
during the neural induction phase, we used an orbital shaker for the OS protocol and a rotating
chamber for the RC protocol. Following the reviewer suggestions, we have now updated our
schematics in **Fig 2D, G**. We now also included the correlation plots showing the representation of
each of the morphological parameters across PC components in the **Fig 2F, I** (as the data would not fit
in main Figure 1).

We rearranged figures EV1/EV2 to favor readability and interpretability of the graphs. The Methods
section was updated accordingly (lines 527-530).

5. Related to this point, the schematic overview of the different protocols in Figures EV1 and EV2
helps to illustrate how they differ in the differentiation process of the organoids. However, it is
unclear from these schematics what the differences are in the aggregation step. Please edit the figures
and the Methods section accordingly to make this more intuitive.

Following the reviewer's suggestion, we edited the figure (**new Fig 2D, G** top panels with new
schematics) and the Methods sections (lines 522-530).

6. In Figure 3B, the PAX6+ cells appear located within only the inner rosettes of the organoids. What
is the identity of the rest of the cells (that is also DCX-). Additionally, more markers should be used to
confirm cortical identity (e.g. EMX2). Is similar patterning observed both the 'PA' and 'OS' protocols?
Please provide comparative stainings (whole organoid and zoom-ins) and including quantifications of
rosette positivity across organoids from both lines used.

We appreciate this important point. We have re-analyzed the images (post-processing) and now
present higher-quality images that clearly visualize PAX6 staining, as well as DCX labeling at the

perimeter of the rosette (**revised Fig 3B**). Regarding the concern about dorsal identity, we hope the
reviewer agrees that PAX6 is a widely recognized marker of cortical progenitors with dorsal identity
as in (Yoon et al., 2019 PMID: 30573846; Li et al., 2023 PMID: 37964131 and others). To further
support the cortical identity, we investigated the presence of a ventral forebrain protein namely DLX2
expressed in interneuron progenitors (**new Fig EV3A**). DLX2 was not expressed. In addition, we
detected PAX6 and DCX in day 20 organoids generated with the OS protocol. No differences were
observed in the expression patterns of these two proteins (**Fig 3B, EV3A**). For both conditions we
now show images of a whole organoid and close-up next to it.

To further characterize the RC compared to the OS organoids we analyzed the patterning over time,
across batches and lines and quantified the expression of proteins expressed in the dorsal forebrain
(**new Fig 3B-D**). We conclude that RC organoids are similar to the OS organoids regarding the
patterning, except that the RC organoids have more proliferative PAX6⁺ progenitors (**new Fig 3C**). In
addition, the expression of the pan-cortical neuronal protein NeuroD2 was comparable across RC and
OS-derived organoids and timepoints (**new Fig 3E**).

PA organoids were excluded from these comparisons for the same reasons raised by the reviewer in
point 3 (see above).

7. Related to the previous point, the positivity (and layering, and co-staining for e.g.
stem/differentiated progeny) for the different markers shown in Figure 3 should be significantly
extended. This should be shown across different organoids and different batches, with associated
quantifications, in order to strengthen the observation of improved reproducibility using the "RC"
protocol. Comparative stainings across the 'PA' and 'OS' protocols (with quantifications) are needed as
well to provide phenotypic comparisons.

As mentioned in point 6, following the reviewer's suggestion we further characterized the RC
organoids patterning over time (day 20, 25, 45 and 90), across batches and lines and compared it to
the OS organoids analyzing PAX6, Ki67 and NeuroD2 (**new Fig 3B-D**). These new findings are
discussed in the Results section (lines 257-284).

PA organoids were excluded from these comparisons for the same reasons raised by the reviewer in
point 3 (see above).

8. RC organoids appear overall bigger (area, EV2), are the stem cells more proliferative?

We thank the reviewer for this important observation. To verify whether the stem cells are more
proliferative, we assessed Ki67 expression in the organoids at day 25, 45 and 90 and quantified the
fraction of progenitors expressing both Ki67⁺ and PAX6⁺ comparing OS and RC protocols. The
analysis revealed an overall larger fraction of double-positive cells in RC organoids, suggesting
higher progenitor proliferative activity, which may contribute to their overall larger size. These results
are now presented in **new Fig 3C** and **new EV3C** and discussed in the manuscript (lines 275-278).

9. The single-cell RNA sequencing data is presented in a too superficial way. While cell classes are
annotated, few markers are shown to substantiate this annotation. An extensive dot plot with markers
for the different cell types should be shown to support their presence (exM, exN, exDP1, etc.).
Consider also integrating with existing previous datasets of the comparative protocols used.

We now provide a list of all markers expressed by the cell types that were annotated (**new Dataset**
**EV5**). In brief, the annotation was performed after integrating the organoid dataset with the human
fetal brain reference dataset (Polioudakis *et al.*, 2019 PMID: 31303374), as described in Methods
(lines 685-693). In addition, we now included a dot plot showing the markers expression for each cell
type in the fetal brain reference (annotation done by the authors of the referenced article) as compared
to the organoids dataset. These data are included in **new Fig. EV4B**.

To further extend our comparison, we analyzed a published dataset using a similar protocol to
generate cortical organoids - "KH" Khan et al., 2020 PMID: 32989314. We processed this dataset in
the same way as the RC dataset, and performed an integration with all organoid cells and the human
fetal brain, to verify whether the cellular diversity obtained with the RC protocol was comparable to
previous similar protocols. We found that both organoid datasets had similar cellular composition to
the fetal brain, with fluctuations being mainly related to the proportions of the different cell types, and

a slight cell line-dependent effect in the KH dataset. We now added these new data in **new Fig.**
**EV4E-G** and discuss the data in the revised manuscript (lines 335-343).

10. After day 25, the organoids generated using the "RC" protocol are moved back on to an orbital
shaker. Why is this choice made? Does the fFSS not influence the organoids at this point? This could
be of interest to show.

This is a good point. The transition from the RC system to an orbital shaker after day 25 was
motivated by both biological and practical considerations. As organoids continue to grow over time,
the confined space within the RC system would eventually lead to physical contact and fusion events,
potentially compromising spatial patterning and reproducibility. Moreover, the RC platform is
recommended for use over a limited time window (up to two weeks), and maintaining long-term
cultures exclusively in this system would be cost-demanding and logistically complex. Importantly,
our data indicate that biomechanical modulation during the early developmental window is sufficient
to stabilize morphogenetic trajectories. However, we agree that prolonged RC culture could represent
an intriguing avenue for future investigation. Future studies should explore whether extending the RC
culture phase may further enhance not only reproducibility but also tissue. We have now clarified this
point in the revised version of the manuscript by adding a new paragraph in the Discussion section
(lines 450-459).

11. Related to the previous point, the differences between the fFSS of the orbital shaker versus the
rotating chamber should be shown.

We have now added this information in the main text of the revised manuscript (lines 138-140).
Specifically, the orbital shaker culture systems have shear stress values ranging from 1.57 to 1.93×10^{-2}
188 Pa corresponding to the maximum and minimum velocities of 0.42 and 7.24×10^{-2} m/s (as discussed
in Saglam-Metiner et al, 2023 PMID: 36788328, Fig. 1). For the rotating-chamber system, the range
of shear stress values is lower, below 14 mPa at 20 rpm, on the suspended spheroids (**Fig 1b**,
Wrzesinski & Fey, 2018 PMID: 29518979).

**Referee #2:**

The manuscript presents an important technical advancement in brain organoid culture, offering a
compelling case that a low-shear rotating chamber improves reproducibility in organoid development.
The work is timely, technically well-executed, and broadly relevant. The conclusions are generally
well-supported, although several aspects of the experimental design and data presentation require
clarification or reframing. The report below highlights specific points that need to be addressed before
the manuscript can be considered for publication.

Major Comments

1. Comparison with Orbital Shaker in CFD Analysis (Figure 1)

The manuscript claims that the RC minimizes fFSS, but Figure 1 only includes CFD simulations for
the RC. Since the RC is compared to an orbital shaker in later sections, it would be beneficial to show
CFD or at least cite data on shear stress levels in an orbital shaker. This would better justify the claim
that RC's low-shear environment is unique.

We have now added this information in the main text of the revised manuscript (lines 138-140).
Specifically, the orbital shaker culture systems have shear stress values ranging from 1.57 to 1.93×10^{-2}
210 Pa corresponding to the maximum and minimum velocities of 0.42 and 7.24×10^{-2} m/s (as discussed
in Saglam-Metiner et al, 2023 PMID: 36788328, Fig. 1). For the rotating-chamber system, the range
of shear stress values is lower, below 14 mPa at 20 rpm, on the suspended spheroids (**Fig 1b**,
Wrzesinski & Fey, 2018 PMID: 29518979).

2. RC vs PA Comparison at Day 6 is Misleading

The RC protocol is compared to the "PA" protocol at day 6, but the RC device is not yet in use at this
stage. The observed morphological differences at this point are due to the aggregation method

(AggreWell vs static), not the rotating chamber. This comparison should be reframed or explicitly
stated to be a comparison of aggregation methods

We thank the reviewer for highlighting this important point. As correctly pointed out by the reviewer,
given the importance of early morphogenetic control on reproducibility (Gjorevski *et al.*, 2022 PMID:
34990240) we also questioned the effect of keeping the organoids on AggreWell for short (1 day) or
longer times (6 days) and therefore checked the possible effects on the morphology of the organoids.

To better clarify that PA EBs were cultured only to test the impact of early aggregation, we updated
the Methods section for the PA protocol (Methods, lines 522-525) and changed the schematic in **Fig**
**2D** (top panel).

Our results suggested that longer aggregation produces more reproducible early EBs (**Fig. 2D**
bottom).

Building on this finding and in agreement with the reviewer's suggestion, we compared the organoids
obtained using the RC and the OS protocol, morphologically (**Fig 2G-I, EV2**), histologically (**new**
**Fig 3, EV3**) and metabolically (**Fig 5**). The protocols are now clarified and better explained.

3. No Data Beyond Day 6 for PA Protocol

The schematic implies PA organoids are cultured to day 90, but only day 6 data are shown. This is
misleading. Either provide later data for the PA protocol or revise the schematic and text to reflect
that the PA comparison is limited to early aggregation.

We have revised our schematics and text to clearly indicate that the comparison with the PA protocols
concerns only the initial aggregation phase (**Fig 2D**, Methods, lines 522-525).

4. Limited Justification for Using RC Only During Days 6-25

The RC is used only during neural induction. No rationale is given for discontinuing RC culture at
242 day 25. Considering its apparent benefits, why not continue its use during maturation? If there are
243 technical limitations, these should be acknowledged.

We thank the reviewer for this question. The transition from the RC system to an orbital shaker after
245 day 25 was motivated by both biological and practical considerations. As organoids continue to grow
over time, the confined space within the RC system would eventually lead to physical contact and
fusion events, potentially compromising spatial patterning and reproducibility. Moreover, the RC
platform is recommended for use over a limited time window (up to two weeks), and maintaining
long-term cultures exclusively in this system would be cost-demanding and logistically complex.
Importantly, our data indicate that biomechanical modulation during the early developmental window
is sufficient to stabilize morphogenetic trajectories. However, we agree that prolonged RC culture
could represent an intriguing avenue for future investigation. Future studies should explore whether
extending the RC culture phase may further enhance not only reproducibility but also tissue. We have
now clarified this point in the revised version of the manuscript by adding a new paragraph in the
Discussion section (lines 450-459).

5. No Marker or Cell Composition Data for Orbital Shaker Protocol

Although the manuscript compares morphological outcomes between RC and orbital shaker protocols,
it does not include scRNA-seq or immunostaining for the orbital shaker condition. Since the claim is
that RC improves reproducibility, it would be valuable to show whether this applies to cellular
composition and tissue identity as well.

Following the reviewer's suggestion, to extend our comparisons to cellular composition and tissue
identity in the RC vs OS-derived organoids, we included new immunostaining characterization and
quantifications to evaluate the differences between the two protocols (**new Fig 3B-F**). We performed
immunostainings of dorsal progenitors and neuron proteins/markers (PAX6 – **new Fig 3B,C**,
NeuroD2 – **new Fig 3E**) across timepoints (day 20, 25, 45, 90), from three batches and two cell lines.
These new data and relative quantifications are now presented in the revised manuscript (lines 257-
284).

We agree that scRNA-seq could also be useful to have as a comparison for the orbital shaker protocol.
However, our claims that required the analysis of scRNA-seq aimed at complementing the
morphological and histological analysis, to ensure that the cellular types detected in the RC-derived

organoids well represent the human fetal brain-derived cells. Future investigation could be performed
to assess more in depth the molecular differences that might be present between RC and OS-derived
organoids. We now mention this possibility in the Discussion (lines 426-428).

6. Gruffi Filtering Requires Justification

The authors use Gruffi to remove stress-affected cells from the scRNA-seq dataset. This is a non-
standard approach. The rationale should be explained, and the impact of this filtering on inter-line
comparisons should be discussed.

We thank the reviewer for this comment. *Gruffi* is a computational tool that identifies and removes
stress-affected cells in an unbiased manner, based on a granular functional filtering approach (Vertesy
*et al.*, 2022 PMID: 35919947). It has been recently shown that brain organoids, unlike fetal brain
tissue, frequently exhibit a distinct cellular stress signature, which can interfere with the accurate
interpretation of lineage trajectories and developmental fidelity. For this reason, *Gruffi* has been
adopted as a quality control step in organoid single-cell transcriptomic analyses, including in datasets
that were generated with a similar organoid protocol. We have now updated the results (lines 313-
314) and Methods sections (lines 677-683) to include relevant recent publications that employ *Gruffi*
as a QC tool to ensure our analytical pipeline aligns with current standards (Kim *et al.*, 2025 PMID:
38559133; Martins-Costa *et al.*, 2023 PMID: 37842725; Dony *et al.*, 2025 PMID: 39951527).

7. Justification of Reference Dataset (Polioudakis et al.)

The authors use a mid-gestation fetal brain dataset (GW17-18) for annotation. They should explain
why this is an appropriate reference for day 90 organoids.

We thank the reviewer for raising this point. We have now clarified in the Methods section the
rationale behind our choice of reference dataset (lines 685-693). Specifically, we selected the GW17-
18 fetal brain dataset (Polioudakis *et al.*, 2019 PMID: 31303374) because it represents a
developmental stage that aligns with day 90 guided cortical organoids (Tanaka *et al.*, 2020 PMID:
32049002). In this article, day 90 organoids were shown to resemble the molecular identity of human
fetal cortex around post-conception week 15-17 (see their Fig 2F), therefore justifying the choice of
this reference dataset.

8. Metabolic Profiling is Interesting but Shallow

The OCR analysis reveals increased OXPHOS in RC organoids, but the metabolic phenotype is not
deeply characterized. No data on ATP, ROS, or glycolysis are presented. These could contextualize
whether elevated OXPHOS reflects better health, more mature neurons, or simply more mitochondria.
At a minimum, the authors should discuss these limitations.

We fully agree that a more comprehensive metabolic characterization of brain organoids, including
direct comparisons with fetal brain tissue, would be of great interest because during
neurodevelopment, neuronal differentiation is accompanied by a metabolic shift from glycolysis
toward OXPHOS (Agostini *et al.*, 2016 PMID: 27058317). While we developed a novel protocol that
enables precise and reliable quantification of mitochondrial oxygen consumption in brain organoids,
our current dataset does not allow us to determine whether the observed increase in OXPHOS reflects
enhanced neuronal maturation, more closely resembling the physiological state, or simply improved
overall organoid health. We believe that such a long-term study is beyond the focus of the present
manuscript.

We now included these raised points in the Discussion of the revised manuscript (lines 446-448).

9. Discrepancy Regarding the Presence of InMGE Cells

In Figure 4B, the scRNA-seq analysis shows a visible population of InMGE cells (interneuron
progenitors from the medial ganglionic eminence) in RC organoids. However, the text states that
ventral forebrain markers were undetectable, reinforcing dorsal identity. This discrepancy should be
clarified. If InMGE cells were identified based on transcriptional similarity, the authors should

acknowledge this and discuss whether these may represent real ventral differentiation events,
misclassification, or noise. Clarification is essential to support the claim of strict dorsal patterning.
We thank the reviewer for raising this concern, we agree with the importance of this clarification.
Following his/her concerns we have subsetted the interneuron clusters of the single cell dataset from
the integrated published fetal brain data and the organoid-derived data (our study). Following
dimensionality reduction and clustering analysis, we noticed that organoid-derived cells mostly do not
integrate with cells of the fetal dataset (**new Fig EV4C**), although they express GABAergic markers
(see GAD1 expression, **new Fig EV4D** top left).
The organoid-derived interneurons do not fully match the MGE identity of the fetal brain dataset, as
they do not express canonical markers of this region (e.g. LHX6, SOX6, SST feature plots in **new Fig**
**EV4D** show expression in the cells that come from the fetal dataset but not as much from the
organoid-derived cells). Therefore, we now renamed the GABAergic cells of the organoid dataset as
“Interneuron cluster 1” and “Interneuron cluster 2”, and updated the presentation of the data in the
manuscript. In the revised manuscript we discuss that the generation of interneuron types could arise
also from dorsal progenitors as reported in recent publications (Delgado *et al.*, 2022 PMID:
34912114; Kim *et al.*, 2023 PMID: 37986891, lines 324-329).

Minor Comments

1. Terminology

Define technical terms like "Dirichlet Normal Energy" in biological terms to aid comprehension.
We have now defined this technical term in the Methods section (lines 591-595).

2. Protocol Definitions. Clarify differences between PA, OS, and RC protocols. Ensure the timeline
and use of dynamic/static conditions are clearly and consistently described.

Thank you for this remark, we have now clarified these points in the revised versions (**Fig 2D** top, **2G**
top and lines 522-530).

3. Data Integration. The authors should mention the possibility of comparing their data with other
published scRNA-seq datasets (e.g., Velasco *et al.*, Øhlenschläger *et al.*, Saglam-Metiner *et al.*) to
contextualize findings.

To contextualize our findings, we analyzed a published dataset using a similar protocol to generate
cortical organoids - “KH” Khan *et al.*, 2020 PMID: 32989314. We processed this dataset in the same
way as the RC dataset, and performed an integration with all organoid cells and the human fetal brain,
to verify whether the cellular diversity obtained with the RC protocol was comparable to previous
similar protocols. We found that both organoid datasets had similar cellular composition to the fetal
brain, with fluctuations being mainly related to the proportions of the different cell types, and a slight
cell line-dependent effect in the KH dataset. We now added these new data in **new Fig. EV4E-G** and
discuss the data in the revised manuscript (lines 335-343).

4. Schematic Clarifications

In figures or text, indicate explicitly that PA organoids were not cultured beyond day 6.
We clarified this point in the schematic of **Fig 2D** top.

**Referee #3:**

In this manuscript, Aiello *et al.* make a case for improving brain organoid generation by minimizing
fluid flow shear stress, which can impact organoid development and consistency. Using a new vertical
rotating chamber (RC) system, they show better reproducibility across batches and iPSC lines, a
metabolic shift toward oxidative phosphorylation, and more controlled neural fate acquisition that
closely models human cortical development. The work adds valuable insight to the field, but a few
important points need to be addressed for the findings to be fully convincing:

1/ A major goal of this article is to generate organoids with a reproducible shape. While this is indeed
a useful goal, a caveat here is that high quality organoid batches are often not round, but with
prominent lobes corresponding to ventricular-like structures that stick out of the organoid. This is

precisely the point of Chiaradia et al, 2023, where it is shown that organoids with more complex
morphologies better mimic in vivo fetal brain development. While this article is cited, this apparent
conflict should be clearly stated. Nevertheless, protocols for organoids with more regular and
therefore reproducible shapes can indeed be needed for certain applications and this does fill a need in
the field. All these considerations should be discussed.

We thank the reviewer for raising this aspect, we have now discussed in more depth the
morphological complexity reported in Chiaradia et al., 2023 where the authors utilized a protocol to
generate organoids based on Matrigel. In our study we used a Matrigel-free guided protocol and we
propose that under these conditions we also have a good correspondence between morphology and
cellular fate acquisition as confirmed by the scRNA-seq data analysis of human fetal brain. With our
work we mainly focused on the reproducibility of the organoid protocol. Following the reviewer's
suggestion these aspects are now clarified in the Discussion (lines 406-409; 465-472).

2/ Related to this, on page 7, the authors state that the organoids maintain a consistent spherical shape
during differentiation. However, by day 45, the organoids appear to exhibit a more irregular
morphology, which contradicts this claim. The authors should address these differences and clarify
whether the observed morphological changes are representative or due to experimental variation.

To avoid any inconsistency, we changed the sentence "the organoids maintained a consistent spherical
shape" to "the organoids maintained a spherical-like shape" (line 166), as the organoids appear to
maintain this shape as visible from the bottom panel of Fig 2A. In addition, to please the reviewer we
have taken the existing data used to generate Fig. 2 and plotted the roundness across the different time
points. As shown in the figure for the reviewer only, a significant difference is detected only between
401 day 6 and all other timepoints, but not among day 25, 45 and 90 timepoints.

**Figure for the reviewer. Quantification of organoid roundness over time.**

Quantification of organoid roundness at different time points from day 6 to day 90. Each dot represents a single
organoid, and bars show mean \pm SEM. Statistical analysis was performed using one-way ANOVA followed by
Tukey's multiple comparisons test. Comparisons involving day 45 are highlighted in red. ns = not significant,
**** p < 0.0001.

3/ The presentation and text for Figure 2 should be simplified and clarified. In figures 2D, E, the
authors compare EB generation methods and analyze results early, at Day 6, while in figure 2F, 2G
they compare agitation methods, in both cases with their EB formation assay, and analyze results at
412 day 90. The text should be clarified, simplified and shortened to clearly describe what the 2 variables
are in these 2 experiments. Also, the OS protocol could be better described.

We thank the reviewer for the suggestion and revised accordingly the text describing Fig 2 (lines 180-
185, 223-226), the schematics of Fig 2D and 2G, and the description of the protocols in the Methods
(lines 522-530).

4/ Aiello et al. state that the RC protocol "preserves both the external morphology and the inner
cytoarchitecture" of organoids. However, the neuroepithelial rosettes, which are highlighted as a key
feature of early neural patterning, are reported to localize predominantly in the innermost regions of
the organoids. Could the authors elaborate on why these structures are not also observed at the
periphery? Is this distribution expected for dorsal forebrain patterning at day 20, or could it indicate

limitations in nutrient diffusion, polarity establishment, or differentiation gradients within the
organoid?

We thank the reviewer for raising this point. We have now included a more extensive characterization
of the rosettes at day 20 and 25. We found that ZO1⁺ rosettes localized in different regions of the
organoids (**new Fig EV3B**), not only in the center (**Fig 3B**). These rosettes have dorsal identity
predominantly, as they are PAX6⁺ (**new Fig 3B**).

We have included these new results in the revised manuscript (lines 269-274).

5/ Hypoxia is often observed at the center of the organoids, and imaging-based studies often therefore
concentrate on the outside regions. Given that here the lumen are concentrated in the center, the
authors should verify the levels of hypoxia and cell death in these regions.

To assess the cell death in the organoids obtained with the RC and OS systems, we performed an
immunostaining against the cell death marker Cleaved caspase 3 (CC3) at day 20 and 25. We then
applied an unbiased, automated analysis pipeline (Methods, lines 638-652) in which, the same
approach used for Sholl analysis, was used to measure the percentage of CC3⁺ cells from the center of
the organoids towards the periphery. These new data are now presented in **new Fig 3F** and show that
RC organoid tend to have a lower percentage of CC3⁺ cells in all areas. We present these results in the
manuscript (lines 285-292).

6/ In Figure 3B, the DAPI staining shows the presence of some PAX6-negative neuroepithelial
rosettes. Could the authors elaborate on the identity of these structures? Are they non-neural, ventral,
or PAX6-negative neural populations? Further clarification would help support the claim of consistent
dorsal forebrain patterning.

We appreciate this important point. We have re-analyzed the images (post-processing) and now
present higher-quality images that clearly visualize PAX6 staining, as well as DCX labeling at the
perimeter of the rosette (**revised Fig 3B**). Regarding the concern about dorsal identity, we hope the
reviewer agrees that PAX6 is a widely recognized marker of cortical progenitors with dorsal identity
as in (Yoon et al., 2019 PMID: 30573846; Li et al., 2023 PMID: 37964131 and others). To further
support the cortical identity, we investigated the presence of a ventral forebrain protein namely DLX2
expressed in interneuron progenitors (**new Fig EV3A**). DLX2 was not expressed. In addition, we
detected PAX6 and DCX in day 20 organoids generated with the OS protocol. No differences were
observed in the expression patterns of these two proteins (**Fig 3B, EV3A**). For both conditions we
now show images of a whole organoid and close-up next to it.

To further characterize the RC compared to the OS organoids we analyzed the patterning over time,
across batches and lines and quantified the expression of proteins expressed in the dorsal forebrain
(**new Fig 3B-D**). We conclude that RC organoids are similar to the OS organoids regarding the
patterning, except that the RC organoids have more proliferative PAX6⁺ progenitors (**new Fig 3C**). In
addition, the expression of the pan-cortical neuronal protein NeuroD2 was comparable across RC and
OS-derived organoids and timepoints (**new Fig 3E**).

7/ Furthermore, to better support the claim of preserved cytoarchitecture, the authors are encouraged
to expand the panel of markers and quantify rosette formation (e.g., using ZO-1 staining for apical
junctions). Figure 3D gives the impression that rosettes are disappearing earlier than with classical
protocols.

Following the reviewer's suggestion we expanded the panel markers to characterize the rosette
formation performing ZO1 staining at day 20, 25, 45 and 90 to follow the rosettes formation and
disappearance, in RC compared to OS-derived organoids. We utilized an automated pipeline for
image analysis to detect and measure the ZO1⁺ area in the organoid sections (Methods) and the
number of rosettes. We have included the data as **new Fig 3D**. The results show that the ZO1⁺ area is
comparable across the two protocols. Both protocols showed a progressive decrease in the number of
rosettes at subsequent timepoints, supporting the natural differentiation of the tissue with an initial
phase of neuroepithelial expansion followed by differentiation into neurons. Interestingly, the total
number of rosettes in the RC system is higher, in line with the RC system promoting a more

proliferative state of the neural progenitors (larger fraction of Ki67⁺ PAX6⁺ progenitors, **new Fig 3C**).
The new data have been integrated in the revised manuscript (lines 270-278).

8/ In Figure 3B, DCX-positive cells are predominantly located at the periphery of the organoids, while
PAX6-positive neuroepithelial rosettes are localized to the innermost regions. If these rosettes
represent neural progenitor zones, it is unclear why newly differentiated immature neurons (DCX⁺)
appear at such a distance. Could the authors clarify whether this spatial separation is consistent with
expected neurogenic progression and radial migration, or if it might suggest a disruption in
organization?

To better support the rosette architecture and demonstrate appropriate fate acquisition, the authors are
encouraged to include additional pan-neuronal markers such as NeuroD2, NeuN or HuC/D alongside
progenitor markers (e.g., Pax6 or Sox2). Co-staining would allow for a more detailed spatial and
temporal analysis of progenitor-to-neuron transition within the organoid.

To address this question, we now include a more extensive characterization of the rosettes at day 25, a
time point of reference used throughout the study. We found rosettes localized in different regions of
the organoids (**new Fig 3B, Fig EV3B**), not only in the center. These rosettes have dorsal identity as
they are populated by PAX6⁺ cells (**new Fig 3B, C**), which show progressive disappearance in both
protocols as time progressed, consistent with progenitor-to-neuron transition within the organoid.

We also include the quantifications of the pan-cortical neuron marker NeuroD2, across RC and OS-
derived organoids and timepoints (**Fig 3E**). Our results suggest a strong similarity between the
organoids derived with the two methods, both of which show the progressive emergence of NeuroD2⁺
cells, in line with the differentiation stage. New data have been integrated in the revised manuscript
(lines 278-284).

Minor points

1/ Could the authors elaborate on the CFD analysis in the results section (page 6)? Similarly, could
the authors provide a scale range for shear forces, to clarify what is here considered as low and high
forces. What is the actual range of shear stress values observed in the RC versus compared systems?
Also, the criteria used to define these thresholds is missing and could help in improving
interpretation.

We have now added this information in the main text of the revised manuscript (lines 138-140).
Specifically, the orbital shaker culture systems have shear stress values ranging from 1.57 to 1.93x 10⁻²
508 Pa corresponding to the maximum and minimum velocities of 0.42 and 7.24 x 10⁻² m/s (as discussed
in Saglam-Metiner et al, 2023 PMID: 36788328, Fig. 1). For the rotating-chamber system, the range
of shear stress values is lower, below 14 mPa at 20 rpm, on the suspended spheroids (**Fig 1b**,
Wrzesinski & Fey, 2018 PMID: 29518979).

2/ The authors claimed that "The shear stress Levels remained low across all tested conditions";
however, the manuscript does not clearly define what these conditions were. Could the authors clarify
this point?

CFD simulations were performed using a Navier-Stokes-based solver to model fluid velocity and
shear stress for constructs with different density related to compactness of the construct affecting
relative weight/buoyancy of constructs with the same surface area (relative mass 1010, 1020, and
1030 where 1000 is density/relative mass of surrounding medium, generated in COMSOL
Multiphysics 6.2 by Resolvent Denmark PS, Maaloev, Denmark). Shear force distribution was
computed based on the chamber's geometry while the velocity profiles were extracted to evaluate the
stability of the system. Data were plotted to compare velocity and shear stress from drag force across
different conditions. We clarified this part in the Methods (lines 532-540).

3/ On page 8, the authors describe RC EBs as both more uniform and exhibiting slightly higher
fluctuations in standard curvature and DNE (Fig EV1E). This appears somehow contradictory, and
some clarification is needed.

We apologize for the confusing phrasing. We now changed this sentence to "The inter-line variability
was further quantified through standard curvature and DNE measurements, which showed strong
morphological reproducibility within lines only for RC protocol, while the PA protocol produced

highly variable values in both lines resulting in no statistically significant difference” (new Fig
EV1C), (lines 215-218).

4/ The authors report the presence of interneuron populations from both the medial (InMGE) and
caudal (InCGE) ganglionic eminences based on scRNA-seq analysis. However, they also state that the
ventral forebrain marker NKX2.1 was not expressed, reinforcing a dorsal forebrain identity. This
presents a potential contradiction, as NKX2.1 is a critical and well-established marker of MGE-
derived interneurons. The authors should clarify how InMGE populations were identified in the
absence of NKX2.1 expression. The authors are encouraged to cite articles identifying dorsally born
interneurons.

We thank the reviewer for raising this concern, we agree with the importance of this clarification.
Following his/her concerns we have subsetted the interneuron clusters of the single cell dataset from
the integrated published fetal brain data and the organoid-derived data (our study). Following
dimensionality reduction and clustering analysis, we noticed that organoid-derived cells mostly do not
integrate with cells of the fetal dataset (new Fig EV4C), although they express GABAergic markers
(see GAD1 expression, new Fig EV4D top left).

The organoid-derived interneurons do not fully match the MGE identity of the fetal brain dataset, as
they do not express canonical markers of this region (e.g. LHX6, SOX6, SST feature plots in new Fig
EV4D show expression in the cells that come from the fetal dataset but not as much from the
organoid-derived cells). Therefore, we now renamed the GABAergic cells of the organoid dataset as
“Interneuron cluster 1” and “Interneuron cluster 2”, and updated the presentation of the data in the
manuscript. In the revised manuscript we discuss that the generation of interneuron types could arise
also from dorsal progenitors as reported in recent publications (Delgado *et al.*, 2022 PMID:
34912114; Kim *et al.*, 2023 PMID: 37986891, lines 324-329).

5/ In the last figure Aiello *et al.* explore mitochondrial metabolism in RC organoids, this is an
important addition to the field. The authors show that scRNA-seq data suggests increased OXPHOS
gene expression in progenitor and radial glial populations, while glycolysis genes are more uniformly
but weakly expressed. However, mRNA levels do not always correlate with metabolic activity. The
authors should acknowledge this limitation and clarify how well these transcriptional signatures are
expected to translate to functional energy states.

We now acknowledge these limitations in line 380.

6/ Finally, the authors discuss the benefits of the RC system over other existing approaches,
particularly mitigating fFSS during early developmental stages. However, it remains unclear why RC
protocol is currently limited to early time points, while organoids are later transferred to conditions
that reintroduce mechanical forces. Given that brain organoids continue to develop and mature over
extended culture periods, the authors are encouraged to discuss the potential for extending the RC
system to later stages. Could this system be adapted or scaled to support prolonged culture while still
maintaining minimal fFSS?

We thank the reviewer for this question. The transition from the RC system to an orbital shaker after
572 day 25 was motivated by both biological and practical considerations. As organoids continue to grow
over time, the confined space within the RC system would eventually lead to physical contact and
fusion events, potentially compromising spatial patterning and reproducibility. Moreover, the RC
platform is recommended for use over a limited time window (up to two weeks), and maintaining
long-term cultures exclusively in this system would be cost-demanding and logistically complex.
Importantly, our data indicate that biomechanical modulation during the early developmental window
is sufficient to stabilize morphogenetic trajectories. However, we agree that prolonged RC culture
could represent an intriguing avenue for future investigation. Future studies should explore whether
extending the RC culture phase may further enhance not only reproducibility but also tissue. We have
now clarified this point in the revised version of the manuscript by adding a new paragraph in the
Discussion section (lines 450-459).

Dear Claudia,

Thank you for the submission of your revised manuscript. We have now received the enclosed reports from the referees and I am happy to say that all support its publication now. Only a few editorial requests will need to be addressed before we can proceed with the official acceptance of your manuscript:

- Your ms has 5 main figures and should thus be published as a short report with combined results and discussion sections. Please either combine these sections or add one more main figure to publish your ms as a full article.
- Please add up to 5 keywords to your ms file.
- The "Data Availability Section" (DAS) should only list data and links to data that have been generate in this study. The re-analysis of published data should be mentioned in the main text and the published papers or Datasets must be cited and listed in the references.
- Please remove the author credits from the ms file. All credits need to be entered during online ms submission.
- The funding info CelVivo Aps should be removed from the Comments box in the system and provided as a separate funder in the entries above.
- All main and EV figures need to be uploaded as individual, high production quality Figure files.
- Fig 3G is called out, but the label is missing in the figure, please correct.
- There are 5 Datasets uploaded; their legends need to be removed from the ms and each legend should be provided in its corresponding Excel file as a separate sheet/tab.
- The movie legends should be removed from the ms and each should be provided separately (e.g. readme.txt, Word) and then zipped up with its corresponding movie file so that we have 2 zip folders, Movie EV1 and Movie EV2.
- The Reagent & Tools table needs to be removed from the ms file and uploaded as a separate file.
- Please upload the Source Data as one folder per figure.
- The SUPPLEMENTARY MATERIAL section should be removed from the ms file.
- Please check Figure 2D, Bottom lane row 2. There appears to be a small highlight figure within the image, which can be seen by adjusting the exposure.

Figure Legends - Comments

- Please note that the exact p values are not provided in the legends of figures 3C, D, F; 5B, EV1 A-C; EV2 A-C; EV3 B, C. Please provide exact values as reasonable.
- Please indicate the statistical test used for data analysis in the legend of figure EV5.
- Please note that information related to n is missing in the legend of figure 5A.

I would like to suggest to modify this sentence in the abstract to:

Reducing fFSS minimizes morphological structure variation and preserves transcriptional signature fidelity across differentiation batches and cell lines.

As the findings should be described in present tense as per journal policy.

EMBO press papers are accompanied online by A) a short (1-2 sentences) summary of the findings and their significance, B) 2-3 bullet points highlighting key results and C) a synopsis image that is exactly 550 pixels wide and 200-600 pixels high (the height is variable). The synopsis image should provide a sketch of the major findings, like a graphical abstract. Please note that text needs to be readable at the final size. Please send us this information along with the final manuscript.

Best wishes,
Esther

Referee #1:

The authors have adequately and thoroughly addressed my initial concerns. The more in-depth phenotypic analyses of the organoids, clarified schematics, and more exhaustive single-cell RNA-sequencing analyses make the manuscript in a much better shape and of broader interest and is now suitable for publication.

Referee #2:

The authors have done an excellent job revising their manuscript and have satisfactorily addressed my previous concerns.

Referee #3:

The authors have thoroughly addressed all the raised points, the manuscript is now suitable for publication.

All editorial and formatting issues were resolved by the authors.

Prof. Claudia Bagni
Department of Fundamental Neurosciences
Université de Lausanne
Rue du Bugnon 9
Lausanne 1005
Switzerland

Dear Claudia,

I am very pleased and happy to accept your manuscript for publication in the next available issue of EMBO reports. Thank you for your contribution to our journal.

If you have any questions, please do not hesitate to contact the Editorial Office. Thank you for your contribution to EMBO Reports. I am looking forward to seeing your study published and thank you for thinking of us !

Best wishes,
Esther
